# Histone methyltransferases MLL2 and SETD1A/B play distinct roles in H3K4me3 deposition during the transition from totipotency to pluripotency

Jingjing Zhang[1,2,5], Qiaoran Sun[1,2,5], Liang Liu[1,2], Shichun Yang[1,2], Xia Zhang[1,2], Yi-Liang Miao [ID][1,2,3,4✉] & Xin Liu [ID][1,2✉]

## Abstract

In early mammalian embryogenesis, a shift from non-canonical histone H3 lysine 4 trimethylation (H3K4me3) linked to transcriptional repression to canonical H3K4me3 indicating active promoters occurs during zygotic genome activation (ZGA). However, the mechanisms and roles of these H3K4me3 states in embryogenesis remain poorly understood. Our research reveals that the histone methyltransferase MLL2 is responsible for installing H3K4me3 (both non-canonical and canonical) in totipotent embryos, while a transition to SETD1A/B-deposited H3K4me3 occurs in pluripotent embryos. Interestingly, MLL2-mediated H3K4me3 operates independently of transcription, fostering a relaxed chromatin state conducive to totipotency rather than directly influencing transcription. Conversely, SETD1A/B-mediated H3K4me3, which depends on transcription, is crucial for facilitating expression of genes essential for pluripotency and pre-implantation development. Our findings highlight the role of the H3K4me3 transition, mediated by an MLL2-to-SETD1A/B relay mechanism, in the regulation of transition from totipotency to pluripotency during early embryogenesis.

**Keywords** H3K4me3; MLL2; SETD1A/B; Zygotic Genome Activation; First Lineage Segregation
**Subject Categories** Chromatin, Transcription & Genomics; Development

## Introduction

Histone 3 lysine 4 trimethylation (H3K4me3) is one of the most evolutionarily conserved epigenetic modifications in eukaryotes, playing a crucial role in transcriptional regulation, development, and disease (Lauberth et al, 2013; Shilatifard, 2012). As a hallmark modification at active gene promoters, the intensity and breadth of H3K4me3 peaks are intricately associated with transcription activity (Benayoun et al, 2014; Chen et al, 2015). Emerging evidence suggests H3K4me3's role in modulating the transcriptional pause-release mechanism (Hu et al, 2023; Wang et al, 2023), though its removal has been shown to minimally affect global transcription under certain conditions (Howe et al, 2017). There are contrasting views on whether H3K4me3 directly influences transcription or instead acts to shield developmental genes from repression by inhibiting PRC2 or DNA methylation (Douillet et al, 2020). Intriguingly, during mouse oogenesis, H3K4me3 accumulates non-canonical patterns in a transcription-independent manner, and these non-canonical H3K4me3 patterns are associated with global transcriptional silencing (Dahl et al, 2016; Hanna et al, 2018; Zhang et al, 2016).

Given the pivotal role of H3K4me3 in transcription and development, understanding the mechanisms regulating its deposition on the genome is crucial. In mammals, H3K4me3 deposition is facilitated by the SET1/MLL complex, comprising various histone lysine methyltransferases (SETD1A, SETD1B, MLL1, and MLL2) alongside several core components (Cenik and Shilatifard, 2021). SETD1A and SETD1B predominantly mediate global H3K4me3 deposition (Clouaire et al, 2012; Sze et al, 2020), whereas MLL1 and MLL2 selectively catalyze H3K4me3 at most active gene promoters (Denissov et al, 2014; Hu et al, 2013a; Wang et al, 2009). The recruitment of SET1/MLL complexes family members to chromatin is achieved through three distinct mechanisms: interaction with RNA Polymerase II (Bae et al, 2020; Muntean et al, 2010), recognition of unmethylated CpG islands via the CXXC motif (Hughes et al, 2020; Wachter et al, 2014; Xu et al, 2018), and engagement with specific histone modifications and variants (Hu et al, 2013b; Zhu et al, 2016). Despite advancements in understanding the recruitment dynamics of SET1/MLL complexes, the precise roles and differential recruitment strategies within the SET1/MLL complexes remain partially understood.

The investigation of early mouse embryogenesis presents a unique window into the dynamic regulatory landscape and functional implications of H3K4me3. The transition from non-canonical H3K4me3 patterns in oocytes to canonical patterns at

[1]Institute of Stem Cell and Regenerative Biology, College of Animal Science and Veterinary Medicine, Huazhong Agricultural University, Wuhan, China. [2]Key Laboratory of Agricultural Animal Genetics, Breeding and Reproduction (Huazhong Agricultural University), Ministry of Education, Wuhan, China. [3]Hubei Hongshan Laboratory, Wuhan, China. [4]Frontiers Science Center for Animal Breeding and Sustainable Production, Huazhong Agricultural University, Ministry of Education, Wuhan, China. [5]These authors contributed equally: Jingjing Zhang, Qiaoran Sun. ✉E-mail: miaoyl@mail.hzau.edu.cn; victorlau@mail.hzau.edu.cn

active gene promoters by the late two-cell stage signifies the intricate role of H3K4me3 in embryonic development (Dahl et al, 2016; Zhang et al, 2016). Previous research indicates that embryos lacking the H3K4 methyltransferases SETD1A and MLL2 are non-viable before E11.5 (Bledau et al, 2014; Glaser et al, 2006). In contrast, embryos with a *Setd1b* deletion survive beyond E11.5, while the viability of *Mll1*-deficient embryos ranges from the 2-cell stage to E14.5, depending on the knockout strategy used (Bledau et al, 2014; Cenik and Shilatifard, 2021). The knockdown of WDR82, integral to the SETD1A/B complex, leads to a more critical condition, halting embryo development before the blastocyst stage (Bi et al, 2011). Similarly, the lack of H3K4me3 demethylases KDM5A and KDM5B impedes embryo progression to the blastocyst stage (Dahl et al, 2016; Liu et al, 2016). This strongly suggests the crucial role of dynamic H3K4me3 regulation in early embryo development. However, the deposition and functional mechanisms of H3K4me3 in early embryo development remain unclear.

Our study delves into the contributions of MLL1, MLL2, SETD1A, and SETD1B to H3K4me3 regulation during early embryonic stages. We discerned that while MLL2 is responsible for the H3K4me3 deposition before and during zygotic genome activation (ZGA), SETD1A and SETD1B subsequently assume this role in a redundant manner following the initiation of ZGA. The MLL2-mediated H3K4me3 is transcription-independent, contrasting with the transcription-dependent H3K4me3 facilitated by SETD1A/B. Alterations in H3K4me3 levels during post-ZGA crucially impact the expression of genes vital for the first cell fate determination and pre-implantation embryonic development, underscoring the indispensable role of transcription-dependent H3K4me3 catalysis by SETD1A/B in the first lineage segregation. Overall, these findings indicate that early embryos manage the vital shift from totipotency to pluripotency by selectively using MLL2 and SETD1A/B to catalyze H3K4me3, underscoring the key role of this transition, enabled by an MLL2 and SETD1A/B relay, in controlling cell potency.

## Results

### H3K4me3 is implemented by MLL2 during pre- and peri-ZGA, while SETD1A/B catalyzes H3K4me3 during post-ZGA

To ascertain the histone lysine methyltransferase responsible for catalyzing H3K4me3 in early embryogenesis, we examined the expression patterns of *Mll1*, *Mll2*, *Setd1a*, and *Setd1b* using the transcriptome and translatome datasets GSE169632 (Zhang et al, 2022) and GSE165782 (Xiong et al, 2022). The results revealed that *Mll2* (also known as *Kmt2b*) and *Setd1a* (also known as *Set1a* and *Kmt2f*) were predominantly expressed in mouse oocytes and early embryos, in contrast to the minimal expression of *Mll1* and *Setd1b* (Appendix Fig. S1A, B). Following this, siRNA targeting *Mll2* or *Setd1a* was microinjected into MII oocytes (Dataset EV1), with subsequent verification of mRNA level knockdown (KD) in late two-cell (Late2C) embryos through RT-qPCR (Fig. 1A; Appendix Fig. S1C,D). Remarkably, *Setd1a* knockdown led to an ~20-fold increase in the expression of structurally homologous *Setd1b*, indicating compensatory mechanisms between SETD1A and

SETD1B (Appendix Fig. S1D). Consequently, we performed a dual knockdown of *Setd1a* and *Setd1b* by injecting the combined siRNA into the MII oocyte, validating the effectiveness of double knockdown at the mRNA level at the Late2C stage (Appendix Fig. S1E). Finally, we employed RNA-seq data to facilitate the validation of mRNA levels and assessed the protein levels of MLL2, SETD1A, and SETD1B at the Late2C and blastocyst stages to confirm the efficacy and sustainability of the knockdown (Appendix Figs. S1F–H, S6C, S10C).

It was observed that the non-canonical H3K4me3 (ncH3K4me3) deposited by MLL2 in mature oocytes transitions to canonical H3K4me3 (cH3K4me3) during the major ZGA stage (Millan-Zambrano et al, 2022). To determine the division of labor between MLL2 and SETD1A/B in this transition, we assessed H3K4me3 levels following *Mll2* or *Setd1a/b* knockdown at the early 2-cell (Early2C), Late2C, and morula stages. Immunostaining revealed that *Mll2* knockdown, but not *Setd1a/b*, resulted in a reduction of the overall levels of H3K4me3 in Early2C and Late2C stages, highlighting MLL2's role in both maintaining ncH3K4me3 and establishing cH3K4me3 during the pre- and peri-ZGA stages (Fig. 1B; Appendix Fig. S2A,B). Conversely, knockdown of *Setd1a/b* instead of *Mll2* resulted in a decrease in the level of cH3K4me3 at the morula stage, signifying the responsibility of SETD1A/B in regulating cH3K4me3 during post-ZGA (Fig. 1B; Appendix Fig. S2A,B).

To further explore the regulation of cH3K4me3 by MLL2 and SETD1A/B, we performed spike-in DNA-normalized H3K4me3 CUT&Tag analysis on Late2C embryos and morulae after the knockdown of *Mll2* or *Setd1a/b* (Appendix Fig. S2C; Dataset EV2). Consistent with immunostaining, CUT&Tag analysis confirmed a decrease in genome-wide cH3K4me3 levels in Late2C embryos following *Mll2* knockdown and in morulae after *Setd1a/b* knockdown (Appendix Fig. S2D–F; Fig. 1C). Next, we divided the cH3K4me3 peaks into Late2C-specific, Morula-specific, and shared peaks (Fig. 1D). For Late2C-specific peaks, a specific reduction was observed in *Mll2* KD Late2C embryos (Fig. 1D,E). Regarding Morula-specific peaks, there was a specific reduction observed in *Setd1a/b* KD morula stage (Fig. 1D,E). However, reductions were observed for shared peaks in both *Mll2* KD Late2C embryos and *Setd1a/b* KD morulae (Fig. 1D,E), indicating a transition in the presence of MLL2 and SETD1A/B at cH3K4me3 peaks from Late2C to morula stage. Collectively, these findings elucidate that MLL2 is essential for catalyzing H3K4me3 prior to and during major ZGA, with a subsequent transition to SETD1A/B regulation after major ZGA.

### Canonical H3K4me3 deposition involves transcription-dependent and transcription-independent mechanisms

To investigate whether the genomic targeting of cH3K4me3 is influenced by transcription, DNA sequence and other epigenetic markers, we conducted a comprehensive analysis that integrated H3K4me3 (GSE73952) (Liu et al, 2016), Pol II (GSE135457) (Liu et al, 2020), H3K27ac (GSE207222) (Wang et al, 2022), H3K9ac (GSE143523) (Yang et al, 2021), H2A.Z (GSE188590) (Liu et al, 2022), and CpG density. Initially, we observed a preference for Late2C-specific and Morula-specific H3K4me3 peaks in regions with low CpG density (Fig. 2A). Genes proximal to Late2C-specific H3K4me3 peaks displayed characteristics indicative of two-cell stage-specific expression, enriched with genes related to histone

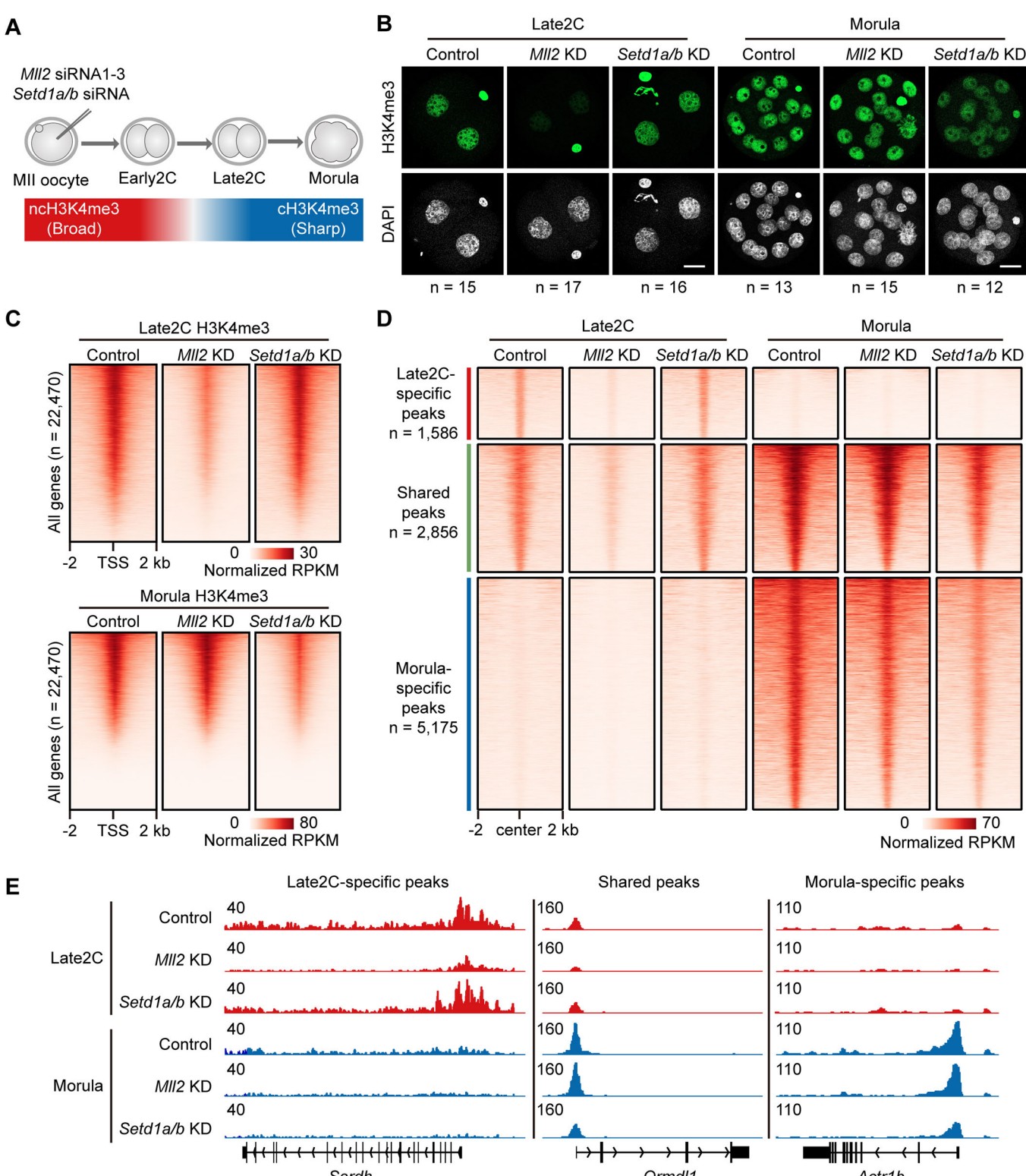

modification, DNA metabolic process (Appendix Fig. S3A; Fig. 2A). Moreover, genes near Morula-specific H3K4me3 peaks exhibited a trend toward morula and blastocyst-specific expression, enriched with genes related to cell adhesion and fate determination (Appendix Fig. S3A; Fig. 2A). Additionally, H3K4me3 peaks in regions of high CpG density within Late2C tended to accumulate further at the morula stage. Genes near these shared peaks showed sustained expression and is especially enriched for housekeeping factors such as translational regulation and ribonucleoprotein complex biogenesis (Appendix Fig. S3A; Fig. 2A).

**Figure 1.** *Mll2* knockdown leads to H3K4me3 reduction in embryos before and during ZGA, whereas *Setd1a/b* knockdown results in H3K4me3 reduction after ZGA.

(A) Diagram detailing the knockdown (KD) strategy for *Mll2* or *Setd1a/b*. This is shown through the schematic representation of microinjecting *Mll2* or *Setd1a/b* siRNA into MII oocytes to effectuate gene knockdown, followed by the initiating in vitro fertilization to generate early embryos. (B) Immunostaining showcases H3K4me3 (green) and DNA (gray) in the Control, *Mll2* knockdown (KD) and *Setd1a/b* KD embryos at the late 2-cell (Late2C), and morula stages. Scale bar, 20 μm. Detailed quantifications are in Appendix Fig. S2B. (C) Heatmaps showing the H3K4me3 enrichment in the Control, *Mll2* KD, and *Setd1a/b* KD at the Late2C (above) and morula (below) stages across all genes (n = 22,470). Each row represents a promoter region (TSS ± 2 kb) and is ordered descending by H3K4me3 enrichment. H3K4me3 enrichment was calculated as normalized reads per kilobase of bin per million mapped reads (RPKM). Two replicates are generated for each sample. (D) Heatmaps showing H3K4me3 enrichment for Control, *Mll2* KD, and *Setd1a/b* KD at the Late2C and morula stages. Each row is classified by Late2C-specific, shared, and morula-specific peaks and are ordered descending by H3K4me3 enrichment (normalized RPKM value). Two replicates are generated for each sample. (E) Genome browser views showing the H3K4me3 enrichment at representative Late2C-specific, shared, and morula-specific peaks. Source data are available online for this figure.

In the Late2C stage, H3K9ac and H2A.Z concurrently localize with H3K4me3 at shared peaks, while H3K27ac specifically co-localizes with H3K4me3 at the morula stage (Appendix Fig. S3B). Comparatively, Pol II and H3K4me3 demonstrate co-localization at both shared and Late2C-specific peaks within Late2C embryos (Fig. 2A). Notably, Pol II engagement with the genome before H3K4me3 at the Early2C stage, indicating Pol II's potential role in facilitating H3K4me3 re-establishment in the Late2C stage. Subsequently, we explored the dependence of H3K4me3 establishment on transcription by using Triptolide to rapidly inhibit transcription initiation. A concentration of 0.1 μM Triptolide effectively inhibited transcription, as determined initially (Appendix Fig. S4A). Late2C and morula stages were treated with 0.1 μM Triptolide for 0, 2, 4, and 6 h (Fig. 2B). The 5-ethynyl uridine (EU) incorporation assay revealed a rapid reduction of transcription in both Late2C and morula stages after 2 h of Triptolide treatment (Appendix Fig. S4B,C). Interestingly, there was a significant reduction in H3K4me3 levels at the morula stage, while H3K4me3 in the Late2C stage showed no significant change (Fig. 2C; Appendix Fig. S4D).

To confirm that the reduction of H3K4me3 was directly due to transcriptional inhibition, we conducted Pol II ULI-NChIP-seq and H3K4me3 CUT&Tag on Late2C embryos and morulae after Triptolide treatment (Dataset EV2). Consistent with EU-staining results, Pol II binding was completely abolished across all transcription start site (TSS) regions after treating Late2C embryos and morulae with Triptolide for 2 h (Appendix Fig. S5). After 2 h of Triptolide treatment, H3K4me3 in morulae exhibited an overall decrease across all TSS regions, maintaining a lower level at 4 and 6 h (Appendix Fig. S5B,C). In contrast, H3K4me3 in the Late2C stage showed no significant changes across all TSS regions (Appendix Fig. S5B,C). On the shared peaks of H3K4me3 between Late2C embryos and morulae, treatment with a transcriptional inhibitor significantly reduced the levels of H3K4me3 in morulae, while the levels of H3K4me3 in Late2C embryos remained unaffected (Fig. 2D). In conclusion, these findings indicate that canonical H3K4me3 at the morula stage is transcription-dependent, whereas in Late2C stage, it is transcription-independent. Additionally, we observed a small fraction (~3%) of H3K4me3 peaks at the morula stage that did not decrease after 6 h of transcriptional inhibitor treatment (Appendix Fig. S5D). These peaks exhibited higher CpG content (Appendix Fig. S5E), suggesting that the targeting of genomic H3K4me3 at the morula stage is also associated with CpG content.

Subsequently, we further explored the potential role of KDM5s in the Triptolide-induced reduction of H3K4me3. To this end, we conducted H3K4me3 immunostaining on morulae treated with Triptolide both alone and in combination with the KDM5s inhibitor CPI-455. Our findings revealed that the co-administration of CPI-455 partially reversed the reduction in H3K4me3 levels observed with Triptolide treatment alone (Appendix Fig. S5F), underscoring the pivotal role of KDM5s in modulating H3K4me3 dynamics in morulae.

## Depletion of ncH3K4me3 and cH3K4me3 has minimal immediate effects on ZGA

We proceeded to investigate the biological significance of MLL2-catalyzed transcription-independent H3K4me3. As H3K4me3 undergoes a shift from a non-canonical to a canonical mode during the transition from minor ZGA to major ZGA, we specifically explore the effects of *Mll2* depletion-induced H3K4me3 reduction on ZGA during Early2C and Late2C stages. Results from EU incorporation assays reveal no significant differences in global transcription levels between control and *Mll2* KD Early2C and Late2C embryos (Fig. 3A; Appendix Fig. S6A). In line with this, embryos from both the control and *Mll2* KD groups exhibit similar gene expression profiles at the Early2C and Late2C stages (Appendix Fig. S6B; Dataset EV2). After *Mll2* knockdown, there are only four and ten differentially expressed genes (FC >2 and P adjusted <0.05) at Early2C and Late2C stages, respectively (Fig. 3B; Dataset EV3). Additionally, significant expression changes are observed in four transposable elements (TEs) only during the Late2C stage (Appendix Fig. S6D). To determine the extent of H3K4me3 contribution to ZGA, we closely examine its correlation with transcriptional changes in the promoters of previously defined ZGA genes (Wang et al, 2022). Despite using a relaxed cutoff to define differentially expressed genes (FC >2 and P adjusted ≤1), we identified only 224 (8%) upregulated genes and 369 (13%) downregulated genes among ZGA genes (Appendix Fig. S6E). Despite the decreased levels of H3K4me3 observed in ZGA gene promoters, there is no alteration in their transcriptional activity, nor a significant correlation with transcriptional changes (Fig. 3C,J; Appendix Fig. S6E). Furthermore, *Mll2* KD embryos exhibited comparable rates of blastocyst formation relative to control embryos and presented no evident morphological abnormalities (Appendix Fig. S6F,G). Finally, we designed three new siRNAs targeting *Mll2* and confirmed consistent results across independent experiments, effectively ruling out off-target effects (Appendix Fig. S7). These findings indicate that the impact of H3K4me3 reduction resulting from *Mll2* knockdown on ZGA is mild.

Considering that overexpression of *Kdm5b* leads to a decrease in ncH3K4me3 and reactivated transcription in oocytes (Zhang et al, 2016),

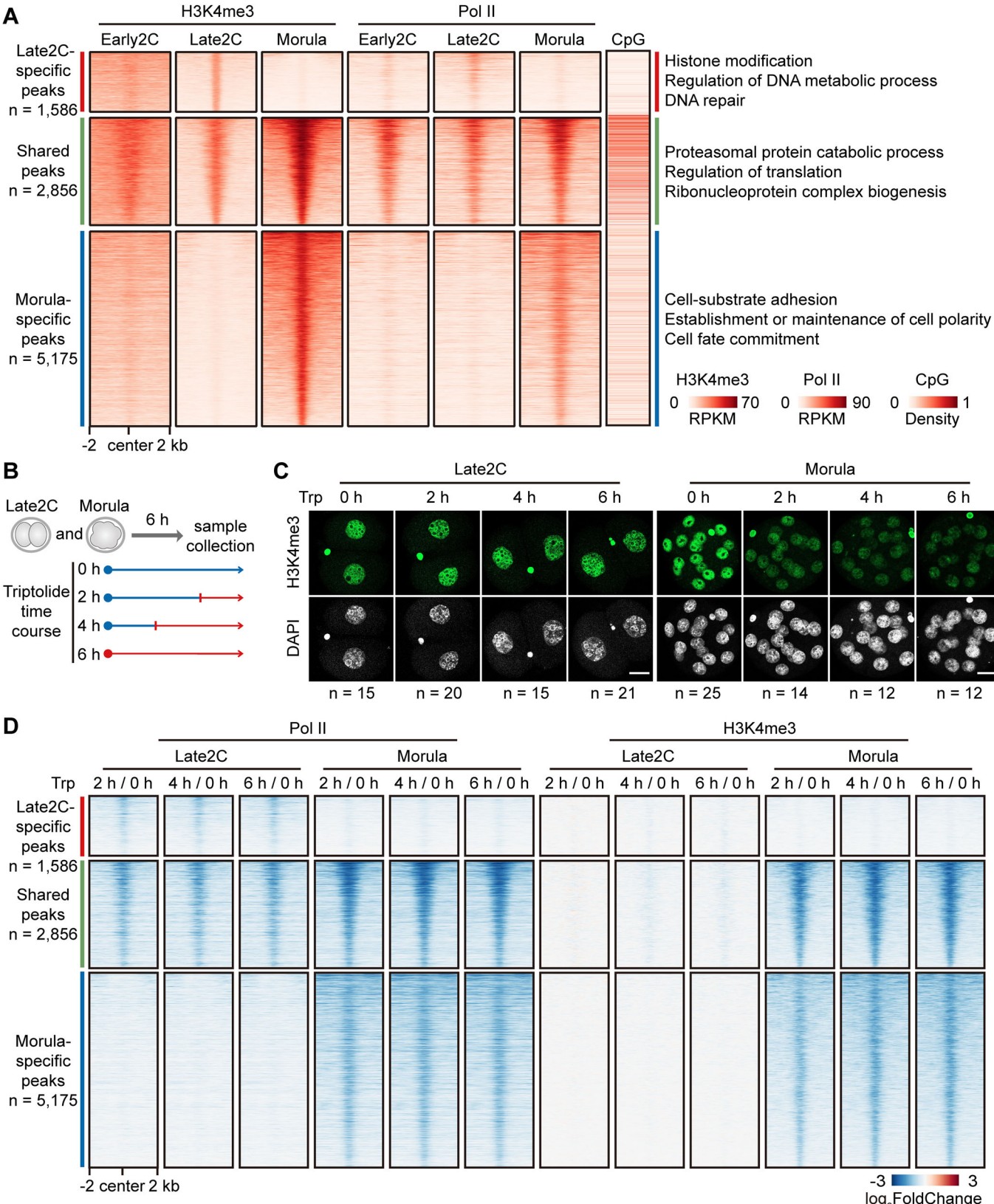

**Figure 2. Canonical H3K4me3 is transcription-dependent in morula but transcription-independent in Late2C.**

(A) Displays heatmaps of H3K4me3, Pol II, and CpG density for Late2C-specific, shared, and morula-specific H3K4me3 peaks, along with Gene Ontology (GO) terms. Rows are sorted by descending H3K4me3 enrichment (RPKM value), with two replicates per sample. (B) Schematic of the triptolide (Trp) treatment protocol for Late2C embryos and morulae, involving a 0.1 μM Trp treatment at 0, 2, 4, and 6 h, followed by collection of both control and treated samples. Treatment time ranges are highlighted in red. (C) Shows immunostaining for H3K4me3 (green) and DNA (gray) in Late2C embryos and morulae following Trp treatment for 0, 2, 4, and 6 h. Detailed quantifications in Appendix Fig. S4D. Scale bar, 20 μm. (D) Heatmap illustrates the changes in Pol II and H3K4me3 enrichment at Late2C-specific, shared, and morula-specific H3K4me3 peaks following Trp treatment, in comparison to the baseline (0 h) over periods of 2, 4, and 6 h. Source data are available online for this figure.

our objective was to investigate whether overexpressing *Kdm5b* could prematurely activate major ZGA. We separately injected wildtype (WT) and catalytically mutated (MUT) *Kdm5b* mRNA into zygotes and then assessed transcription levels at the Early2C stage before major ZGA (Fig. 3D; Appendix Fig. S8A). *Kdm5b* WT expression resulted in a substantial decrease in global H3K4me3 levels at the Early2C stage compared to the control (Fig. 3E; Appendix Fig. S8B). Furthermore, our CUT&Tag analysis revealed that *Kdm5b* overexpression in Early2C embryos led to extensive erasure of the widespread non-canonical H3K4me3 peaks observed in oocytes (Appendix Fig. S8C; Fig. 3F; Dataset EV2). However, this did not result in the premature emergence of canonical H3K4me3 formation in Early2C embryos (Appendix Fig. S8D). It is noteworthy that global transcriptional activity remained unaffected by ectopic expression of *Kdm5b*, as revealed by EU incorporation assay and transcriptome sequencing (Fig. 3G,H; Appendix Fig. S8E,F; Datasets EV2, 4). Specifically for major ZGA genes, *Kdm5b* overexpression significantly reduced non-canonical H3K4me3 at gene promoters, without an increase in ZGA gene expression (Fig. 3I,J; Appendix Fig. S8G). In conclusion, overexpression of *Kdm5b* in pre-ZGA embryos leads to widespread erasure of non-canonical H3K4me3 but does not prematurely activate major ZGA.

## Insufficient impact on ZGA when ncH3K4me3 and cH3K4me3 elevate at the Late2C stage

On the other hand, previous studies have indicated that knocking down *Kdm5a* and *Kdm5b* can result in compromised developmental and failure to reach the blastocysts stage (Dahl et al, 2016). Our investigation aimed to assess whether simultaneous knockdown of *Kdm5a/b* would impair ZGA through ncH3K4me3. Following *Kdm5b* knockdown (Appendix Figs. S9A,B,I and S13G), additional knock-down of *Kdm5a* did not lead to a further elevation of H3K4me3 levels in Late2C embryos, indicating that KDM5B primarily functions as the H3K4me3 demethylase during major ZGA (Appendix Fig. S9C). Consequently, we performed zygotes with the KDM5s inhibitor CPI-455 at the optimal concentration of 25 μM (Fig. 4A; Appendix Fig. S9D). Embryos reducing *Kdm5b* or with inhibited catalytic activity maintained high levels of H3K4me3 at the Late2C stage (Fig. 4B; Appendix Fig. S9E). Next, we performed H3K4me3 CUT&Tag assays for Late2C embryos subjected to *Kdm5b* KD, CPI-455 treatment, and their respective control conditions (Appendix Fig. S9F; Dataset EV2). The genome-wide distribution of H3K4me3 assessed using 5-kb running window significantly increased in Late2C embryos following *Kdm5b* KD and CPI-455 treatment, consistent with the observations made by immunostaining (Fig. 4C). Notably, *Kdm5b* KD and CPI-455 treatment in zygotes had dual effects: abnormal retention of ncH3K4me3 and a significant increase in cH3K4me3 in Late2C embryos (Fig. 4D).

Subsequently, we investigated the transcriptional changes induced by perturbing KDM5B. Initially, we assessed global transcriptional activity in Late2C embryos through EU incorporation assay and observed that it remained unaffected by both *Kdm5b* KD and CPI-455 treatment (Fig. 4E; Appendix Fig. S9G). Consistently, *Kdm5b* KD and CPI-455 treatment exhibited similar gene expression profiles during the Late2C stage, with only 273 and 145 downregulated genes, respectively (Appendix Fig. S9H; Fig. 4F; Datasets EV2, 5, 6). In line with minor gene expression changes, the expression levels of the majority of TEs remained comparable, including MERVL-int and MT2_Mm, which are highly expressed during major ZGA (Appendix Fig. S9J). Given that broad ncH3K4me3 is correlated with transcriptional repression, and narrow cH3K4me3 is associated with transcriptional activation, we aimed to investigate whether changes in H3K4me3 are linked to alterations in gene expression. Specifically, we examined the expression changes of two categories of genes during the transition from MII to Late2C embryos: those undergoing the loss of ncH3K4me3 peaks (C1) and those acquiring cH3K4me3 peaks (C2) at gene promoters (Fig. 4G). Following *Kdm5b* KD and CPI-455 treatment, there was an elevation in H3K4me3 levels at the promoters of C1 and C2 genes (Fig. 4G). However, this increase in H3K4me3 at gene promoters did not directly impact gene expression (Fig. 4H,I). In essence, perturbations in KDM5B resulted in the abnormal retention of ncH3K4me3 and an increase in cH3K4me3 levels during major ZGA, without a direct influence on gene expression.

## SETD1A/B-catalyzed transcription-dependent H3K4me3 is essential for the first cell fate decision

Subsequently, we explored the impact of SETD1A/B-catalyzed transcription-dependent H3K4me3 on early embryonic development. *Setd1a/b* knockdown resulted in the majority of embryos arresting at the morula stage, with only 31% progressing to the blastocyst stage even after extended culture (Fig. 5A,B). To assess the potential influence of SETD1A/B-mediated H3K4me3 on gene expression and development, we initially evaluated global transcriptional activity in morulae through EU incorporation assays. *Setd1a/b* knockdown embryos exhibited a decrease in global transcriptional activity compared to the control (Fig. 5C; Appendix Fig. S10A). Consistent with this, our transcriptome analysis indicated significant differences in gene expression between *Setd1a/b* KD and control groups, with more downregulated genes than upregulated genes in *Setd1a/b* KD morulae and blastocysts (Appendix Fig. S10B; Fig. 5D; Dataset EV2,7). Downregulated genes in morulae and blastocysts were commonly enriched in carbohydrate derivative metabolism and intracellular membrane, while specific Gene Ontology terms were not identified among the

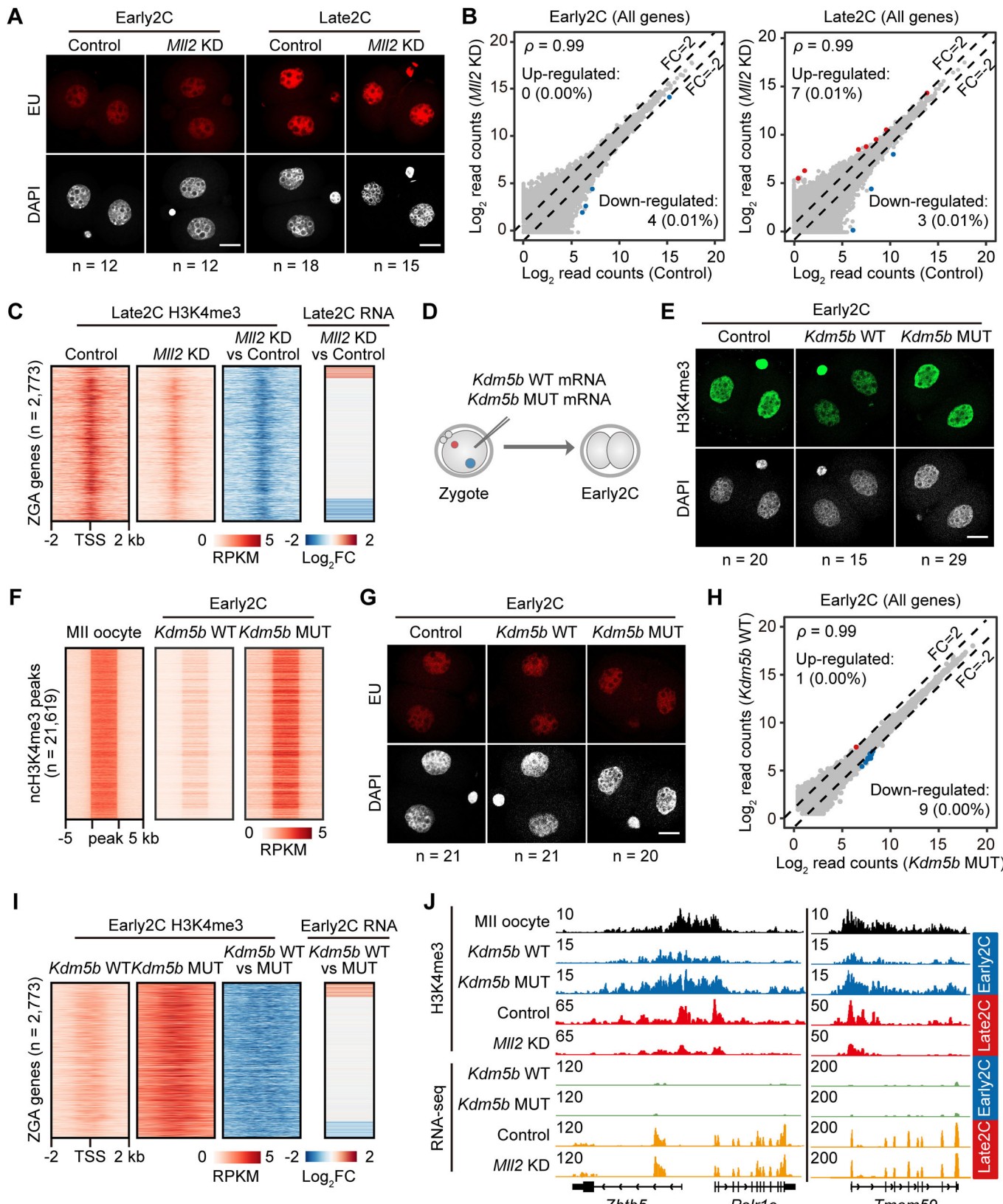

◀ **Figure 3.  Depletion of H3K4me3 has minimal impact on ZGA.**

(A) EU-staining assay reveals global transcriptional activity in Early2C and Late2C embryos. Detailed quantifications are in Appendix Fig. S6A. Scale bar, 20 μm. (B) Scatter plots comparing the gene expression levels between Control and *Mll2* KD embryos at the Early2C (left) and Late2C (right) stages, highlighting genes upregulated in *Mll2* KD embryos (red) and downregulated in Control embryos (blue). Spearman's correlation coefficients are displayed in the top-left. (C) Heatmaps illustrate the differences in H3K4me3 enrichment and RNA expression for ZGA gene promoters ($n = 2773$) between Control and *Mll2* KD at the Late2C stage. (D) Schematic of the microinjection experiments with *Kdm5b* wildtype (WT) and mutant (MUT) mRNA from zygote to Early2C embryos. (E) Immunostaining of H3K4me3 (green) and DNA (gray) in Early2C embryos injected with *Kdm5b* WT and MUT mRNAs. Detailed quantifications in Appendix Fig. S8B. Scale bar, 20 μm. (F) Heatmap showing H3K4me3 enrichment levels within ncH3K4me3 peaks ($n = 21{,}619$) in MII oocytes and Early2C embryos, comparing *Kdm5b* WT to MUT. The characterization of these ncH3K4me3 peaks were based on the ChIP-seq dataset GSE73952 (Liu et al, 2016) derived from MII oocytes. (G) EU-staining assay showing the global transcriptional activity of Control, *Kdm5b* WT, and MUT mRNA overexpressing in Early2C embryos. Detailed quantifications are in Appendix Fig. S8E. Scale bar, 20 μm. (H) Scatter plots comparing the gene expression levels between *Kdm5b* WT and *Kdm5b* MUT embryos at the Early2C stage, highlighting genes upregulated in *Kdm5b* WT embryos (red) and downregulated in *Kdm5b* MUT embryos (blue). Spearman's correlation coefficients are shown in the top-left panel. (I) Heatmaps illustrate differences in H3K4me3 enrichment and RNA expression for ZGA gene promoters ($n = 2773$) between *Kdm5b* WT and *Kdm5b* MUT at the Early2C stage. (J) Genome browser view showing the H3K4me3 enrichments and RNA levels in a representative ZGA gene of Control, *Kdm5b* WT, and *Kdm5b* MUT mRNA overexpressing in Early2C and Late2C embryos. Source data are available online for this figure.

upregulated genes (Appendix Fig. S10D). Subsequently, we sought to investigate whether *Setd1a/b* knockdown affects the first lineage differentiation. Utilizing previously published transcriptome data (Wang et al, 2018), we identified genes that were specifically expressed in the inner cell mass (ICM) and trophectoderm (TE) (FC >2 and FPKM >5). Remarkably, we observed that *Setd1a/b* knockdown resulted in the downregulation of ICM and TE-specific gene expression in morulae and blastocysts (Fig. 5D,E). This included key genes involved in the first cell fate decision, such as *Nanog*, *Pou5f1*, *Tead4*, and *Cdx2* (Fig. 5D). Moreover, both NANOG and CDX2 protein levels were reduced in *Setd1a/b* KD morulae and blastocysts (Fig. 5F; Appendix Fig. S10E). These findings suggest that the reduction of *Setd1a/b* leads to the downregulation of genes essential for the cell fate commitment.

To determine whether the knockdown of *Setd1a/b* regulates gene expression through H3K4me3, we analyzed H3K4me3 enrichment on the promoters of differentially expressed genes in normally developing morulae and blastocysts (ICM and TE). Notably, there was a significant enrichment of H3K4me3 on the promoters of downregulated genes compared to upregulated gene promoters (Fig. 5G; Appendix Fig. S10F). Furthermore, H3K4me3 levels on the promoters of downregulated genes in *Setd1a/b* KD morulae show a marked decrease (Fig. 5G; Appendix Fig. S10G), indicating that the downregulation of gene expression is attributed to the reduction of H3K4me3. Additionally, *Setd1a/b* knockdown led to a significant downregulation of genes near Morula-specific and shared H3K4me3 peaks, while genes near Late2C-specific H3K4me3 peaks did not show significant changes (Appendix Fig. S10H). Subsequently, we conducted rescue experiments in *Setd1a/b* KD embryos by co-injecting either WT or catalytically inactive MUT human *SETD1A* and *STED1B* mRNA (Fig. 5H; Appendix Fig. S11A,B). Successful rescue of developmental defects in *Setd1a/b* KD embryos was achieved only through the injection of WT *SETD1A/B* mRNA, highlighting the critical role of SETD1A/B's methyltransferase activity in early embryonic development (Fig. 5I,J). In summary, these findings suggest that SETD1A/B-catalyzed transcription-dependent H3K4me3 ensures pre-implantation development by promoting the expression of genes essential for the first cell fate decision.

## Demethylase activity of KDM5B is crucial for embryonic development after the Late2C stage

In the *Kdm5b* knockdown, embryos failed to progress to the blastocyst stage (Appendix Fig. S12A,B), prompting our hypothesis that KDM5B regulates the expression of genes essential for cell fate decision in pre-implantation embryos by erasing H3K4me3 marks. Knockdown of *Kdm5b* resulted in elevated levels of H3K4me3 at the blastocyst stage and increased expression of key genes involved in the first cell fate decision, such as *Nanog*, *Pou5f1*, and *Tead4* (Appendix Fig. S12C,D). Immunostaining revealed a reduction in cell numbers and a twofold increase in the percentage of cells expressing NANOG in *Kdm5b* KD blastocysts, with abnormal co-expression of NANOG and CDX2 in the outer cells (Appendix Fig. S12E,F). These results suggest that the proper execution of the first lineage specification events is impaired in the knockdown of *Kdm5b*.

To assess whether KDM5B governs the initial cell fate decision through its enzymatic activity, we conducted rescue experiments by co-expressing WT or MUT *Kdm5b* mRNA in *Kdm5b* KD embryos (Appendix Fig. S13A). Only injection of WT *Kdm5b* mRNA successfully rescued the developmental defects observed in *Kdm5b* KD embryos (Appendix Fig. S13B–D), indicating that the impact of *Kdm5b* KD on early embryonic development is dependent on its enzymatic activity. Additionally, temporal treatment with the CPI-455 determined a critical time window for the catalytic function of KDM5B during early embryonic development (Fig. 6A). Remarkably, treatment with CPI-455 from the Late2C to the blastocyst stage led to decreased developmental rates and significantly increased expression of *Nanog* and *Pou5f1* similar to full-length CPI-455 treatment (Fig. 6B–D).

To determine whether *Kdm5b* KD regulates gene expression through H3K4me3, we first performed H3K4me3 CUT&Tag and RNA-seq analyses in control and *Kdm5b* KD morula samples (Appendix Fig. S13E,F; Dataset EV2). Our findings demonstrate that *Kdm5b* KD significantly elevated H3K4me3 enrichment at gene promoters and broadened the H3K4me3 peaks (Fig. 6E). Furthermore, *Kdm5b* KD resulted in the upregulation of 264 genes and downregulation of 241 genes (Fig. 6F; Dataset EV8). Gene Ontology analysis revealed that upregulated genes predominantly concentrated in the regulation of epithelial to mesenchymal transition and embryonic foregut morphogenesis; the down-regulated genes were largely associated with placental development, aligning with the observed phenotype of diminished TE cell counts in blastocysts (Appendix Figs. S13H, S12F). Additionally, genes such as *Pou5f1*, *Igfbp3*, and *Cxcl14*, which were upregulated following *Kdm5b* KD, showed increased H3K4me3 enrichment and broader peaks at their promoters (Fig. 6G,H). In conclusion, our investigations reveal that KDM5B is pivotal in controlling the

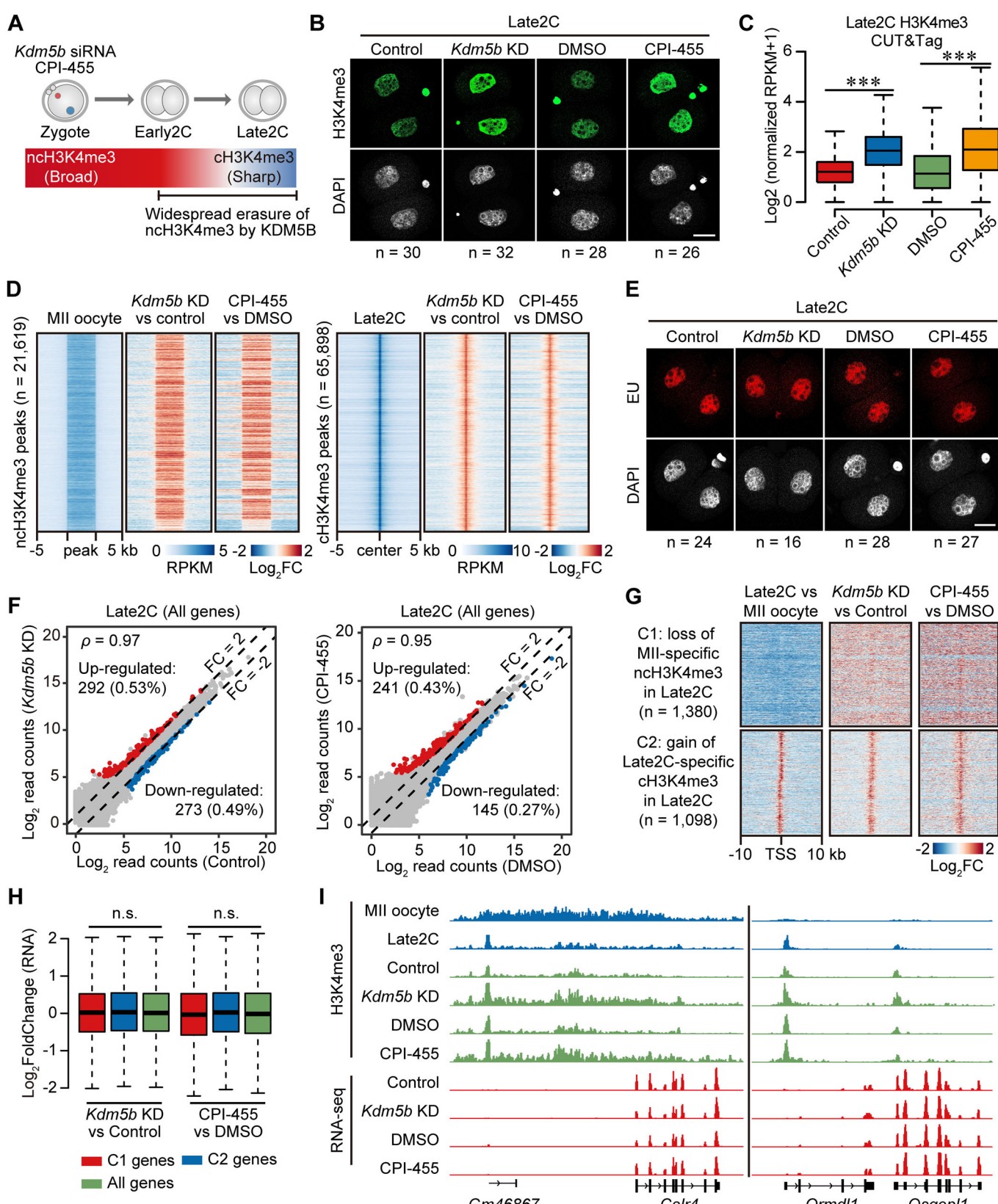

◄ **Figure 4. Retained ncH3K4me3 and increased cH3K4me3 without direct gene expression impact.**

(A) Schematic representation delineates the experimental procedure involving siRNA-mediated knockdown of *Kdm5b* and the application of the KDM5 inhibitor CPI-455 in zygotes. ncH3K4me3: non-canonical H3K4me3; cH3K4me3: canonical H3K4me3. (B) Immunostaining of H3K4me3 (green) and DNA (gray) in Control, *Kdm5b* KD, DMSO-treated, and CPI-455-treated embryos at the Late2C stage. Detailed quantifications are in Appendix Fig. S9E. Scale bar, 20 μm. (C) Box plots showing the H3K4me3 enrichment levels in Control, *Kdm5b* KD, DMSO-treated, and CPI-455-treated embryos at the Late2C stage. H3K4me3 enrichment was calculated as normalized RPKM using 5-kb bins ($n = 546,206$). Boxplot was used to display data distribution, with the median as the central line, the box showing the interquartile range (IQR) from the 25th to 75th percentile, and whiskers extending to data points within 1.5 times the IQR. ***$P < 0.001$; two-sided Wilcoxon–Mann–Whitney test (Control vs *Kdm5b* KD $P < 2.2E{-}16$; DMSO-treated vs CPI-455-treated $P < 2.2E{-}16$). (D) Heatmaps showcase the differential enrichment of H3K4me3 between Control and *Kdm5b* KD embryos, and between DMSO-treated and CPI-455-treated embryos at the Late2C stage, for both ncH3K4me3 ($n = 21,619$) and cH3K4me3 ($n = 65,898$) peaks. ncH3K4me3 peaks were identified in GSE73952 ChIP-seq dataset from MII oocytes, while cH3K4me3 peaks were from Late2C embryos (Liu et al, 2016). (E) EU-staining assay showing the global transcriptional activity in Control, *Kdm5b* KD, DMSO-treated, and CPI-455-treated embryos at the Late2C stage. Detailed quantifications are in Appendix Fig. S9G. Scale bar, 20 μm. (F) Scatter plots comparing the gene expression levels between Control and *Kdm5b* KD embryos at the Late2C stage (left), highlighting genes upregulated in *Kdm5b* KD embryos (red) and downregulated in Control embryos (blue). Similarly, plots for DMSO-treated versus CPI-455-treated embryos at the Late2C stage (right) show genes upregulated in CPI-455-treated embryos (red) and downregulated in DMSO-treated embryos (blue). Spearman's correlation coefficients are displayed in the top-left. (G) Heatmaps reveal the changes in H3K4me3 enrichment at gene promoters near either the loss of ncH3K4me3 peaks (C1, $n = 1380$) or the gain of cH3K4me3 peaks (C2, $n = 1098$) in Late2C embryos. Each row represents a promoter region (TSS ± 10 kb). (H) Box plots showing the expression changes for genes near C1 peaks ($n = 1380$), genes near C2 peaks ($n = 1098$), and all genes ($n = 22,470$) within promoter regions in Late2C embryos, comparing Control versus *Kdm5b* KD, and DMSO-treated versus CPI-455-treated groups. Boxplot was used to display data distribution, with the median as the central line, the box showing the IQR from the 25th to 75th percentile, and whiskers extending to data points within 1.5 times the IQR. n.s., no significance; two-sided Wilcoxon–Mann–Whitney test (*Kdm5b* KD vs Control: C1 genes vs all genes $P = 0.25$, C2 genes vs all genes $P = 0.74$; CPI-455 vs DMSO: C1 genes vs all genes $P = 0.93$, C2 genes vs all genes $P = 0.87$). (I) Genome browser views showing the H3K4me3 enrichments and RNA levels for a representative gene close to C1 and C2 clusters. Source data are available online for this figure.

expression of genes essential for early cell fate determination, through the meticulous modulation of H3K4me3 dynamics.

## Discussion

In early mammalian embryos, non-canonical H3K4me3 associated with transcriptional silencing appears in embryos before ZGA, subsequently transitioning into canonical H3K4me3 on active gene promoters during ZGA (Bu et al, 2022; Dahl et al, 2016; Xia et al, 2019; Zhang et al, 2016). However, how these two distinct patterns of H3K4me3 are established and function in early embryos remains elusive. In our study, we discovered that MLL2 deposits ncH3K4me3 independently of transcription before ZGA, consistent with previous findings in mouse oocytes (Hanna et al, 2018). MLL2 also deposits cH3K4me3 transcription independently during ZGA. Notably, disrupting MLL2-mediated H3K4me3 deposition does not disturb the transcription of ZGA genes. Conversely, SETD1A/B is found to play a role in establishing and maintaining cH3K4me3 after ZGA. This transcription-dependent H3K4me3 is crucial for the expression of critical genes that influence the first cell-fate decision and the development of pre-implantation embryos (Fig. 7).

Previous research on H3K4me3 in mouse oocytes and early embryos has predominantly relied on conditional or developmental knockout strategies targeting SET1/MLL complexes. The conditional knockout of *Mll2*, *Setd1b*, and *Cxxc1* in oocytes notably impacted the quality of oocytes (Andreu-Vieyra et al, 2010; Brici et al, 2017; Sha et al, 2020; Yu et al, 2017), complicating the assessment of their direct effects on early embryonic development versus secondary consequences due to oocyte defects. Developmental knockouts may obscure earlier developmental needs by the persistence of maternal mRNA and proteins in oocytes. For instance, while maternal MLL2, SETD1B, and CXXC1 are essential for oocyte maturation and zygotic development (Andreu-Vieyra et al, 2010; Brici et al, 2017; Yu et al, 2017), embryos deficient in *Mll2*, *Setd1b*, or *Cxxc1* at the zygote stage are still capable of developing to the blastocyst stage (Bledau et al, 2014; Carlone and Skalnik, 2001; Glaser et al, 2006). This raises questions about the

necessity of SET1/MLL complex for early embryonic development in mice. To investigate the role of SET1/MLL-catalyzed H3K4me3 in early embryonic development more directly, we conducted gene knockdown experiments in MII oocytes, thereby ensuring effective gene suppression during this critical developmental phase.

Preliminary immunostaining studies have shown that either the knockout of *Mll2* or overexpression of *Kdm5b* leads to reduced levels of ncH3K4me3, which disrupts genomic silencing in mature oocytes (Andreu-Vieyra et al, 2010; Zhang et al, 2016). These results establish a correlation between ncH3K4me3 in mouse oocytes and pre-ZGA embryos and the regulation of transcriptional silencing in mouse oocytes and pre-ZGA embryos. Interestingly, we found that MLL2 deposits ncH3K4me3 in embryos pre-ZGA and cH3K4me3 during ZGA, with neither directly influencing ZGA gene transcription. In ESCs, MLL2 establishes H3K4me3 at the promoters of developmental genes, protecting them from repression by PRC2-derived H3K27me3 or DNA methylation, rather than directly regulating transcription (Douillet et al, 2020). Similarly, in mature mouse oocytes, ncH3K4me3 is predominantly observed in regions of DNA hypomethylation and exhibits mutual exclusivity with H3K27me3 (Xu et al, 2019; Zhang et al, 2016; Zheng et al, 2016). This suggests that MLL2's role in establishing H3K4me3 may indirectly affect transcription by modulating other epigenetic markers, such as H3K27me3 and DNA methylation, rather than through direct transcriptional regulation. The extensive loss of H3K27me3 and DNA methylation in early mouse embryos (Liu et al, 2016; Smith et al, 2012; Zheng et al, 2016) may explain why reductions in MLL2-mediated H3K4me3 do not influence ZGA gene transcription. Moreover, ncH3K4me3 supports the de novo formation of nuclear lamina-associated domains in zygotes, which do not functionally influence gene expression, unlike the domains established by H3K9me2/3 (Borsos et al, 2019; Guerreiro et al, 2023). Thus, in totipotent embryos before and during ZGA, MLL2-mediated H3K4me3 does not instruct transcription but collaborates with other epigenetic modifications to foster a relaxed chromatin state conducive to totipotency establishment.

During the morula stage before pluripotent embryo formation, there is a transition in H3K4me3 deposition from MLL2 to SETD1A/B. Previous research has shown that the individual

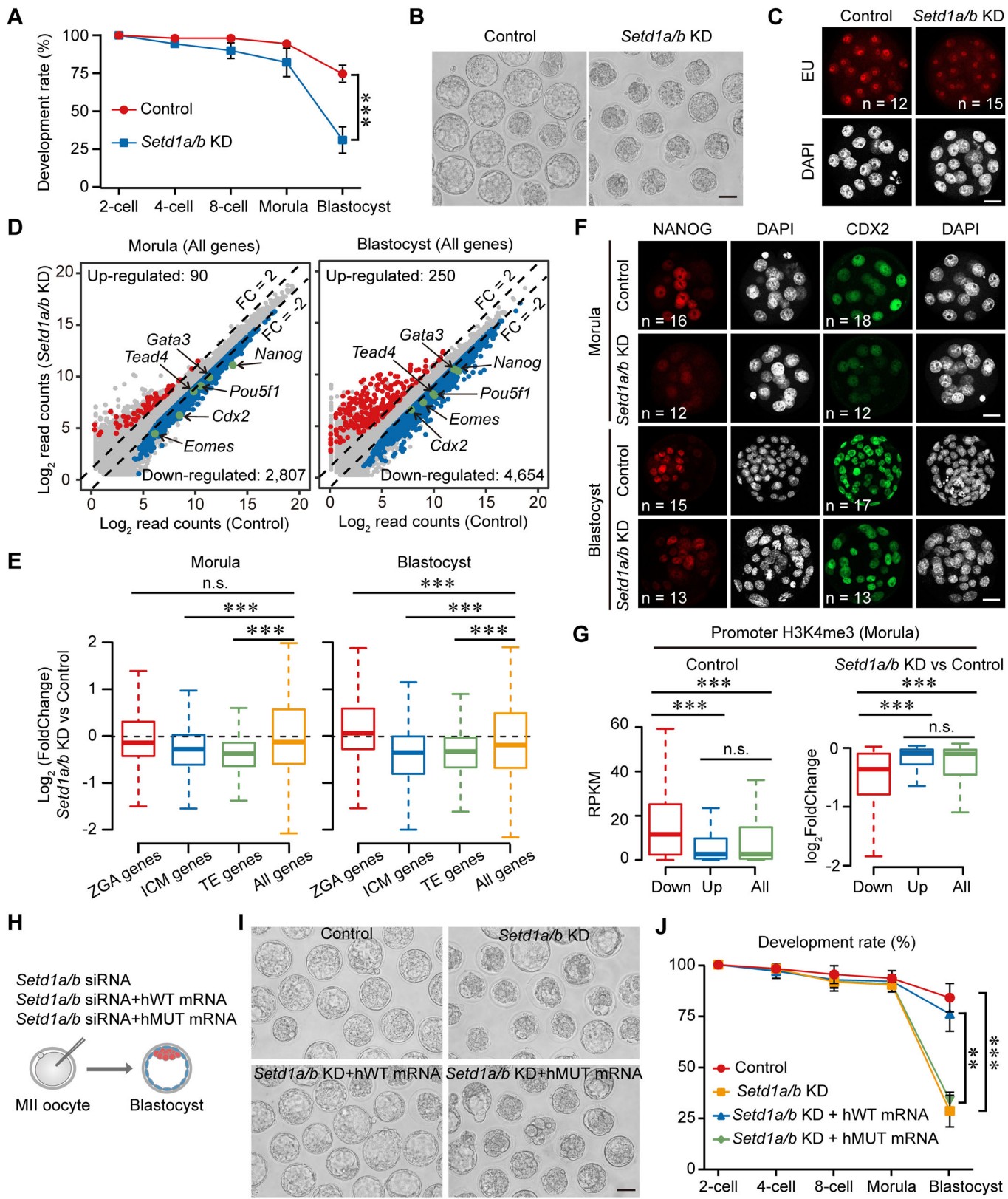

**Figure 5. Knockdown of *Setd1a/b* reduces post-ZGA H3K4me3 levels, resulting in early embryonic arrest.**

(A) Line graphs illustrate the development rates of Control and *Setd1a/b* KD embryos at specific time points. Error bars represent mean ± standard deviation (SD) from three biological replicates. ***$P < 0.001$; two-sided unpaired Student's *t*-test ($P = 0.000075$). (B) Representative images of Control and *Setd1a/b* KD groups at the blastocyst stage. One representative image from three independent experiments is shown. Scale bar, 50 μm. (C) EU-staining assay showing global transcriptional activity in Control and *Setd1a/b* KD morulae. Detailed quantifications are in Appendix Fig. S10A. Scale bar, 20 μm. (D) Scatter plots comparing the gene expression levels between Control and *Setd1a/b* KD embryos at the morula (top) and blastocyst (bottom) stages, highlighting genes upregulated in *Setd1a/b* KD embryos (red) and downregulated in Control embryos (blue). Representative downregulated genes are colored green. (E) Box plots showing the expression changes for ZGA genes ($n = 2773$), inner cell mass genes (ICM, $n = 632$), trophectoderm genes (TE, $n = 697$), and all genes ($n = 22{,}470$) in Control and *Setd1a/b* KD embryos at the morula and blastocyst stages. Boxplot was used to display data distribution, with the median as the central line, the box showing the IQR from the 25th to 75th percentile, and whiskers extending to data points within 1.5 times the IQR. n.s., not significant; ***$P < 0.001$; two-sided Wilcoxon–Mann–Whitney test (Morula: ZGA genes vs all genes $P = 0.94$, ICM genes vs all genes $P = 3.1\text{E-}12$, TE genes vs all genes $P < 2.2\text{E-}16$; Blastocyst: ZGA genes vs all genes $P < 2.2\text{E-}16$, ICM genes vs all genes $P < 2.2\text{E-}16$, TE genes vs all genes $P = 6.7\text{E-}15$). (F) Immunostaining for NANOG (red), CDX2 (green), and DNA (gray) in Control and *Setd1a/b* KD embryos at the morula and blastocyst stages. Detailed quantifications in Appendix Fig. S10E. Scale bar, 20 μm. (G) Box plots showing the H3K4me3 enrichment levels at the promoter regions (TSS ± 2 kb) of downregulated genes ($n = 2807$), upregulated genes ($n = 90$), and all genes ($n = 22{,}470$) when comparing Control with *Setd1a/b* KD groups. Boxplot was used to display data distribution, with the median as the central line, the box showing the IQR from the 25th to 75th percentile, and whiskers extending to data points within 1.5 times the IQR. n.s., not significant; ***$P < 0.001$; two-sided Wilcoxon–Mann–Whitney test (Control: downregulated genes vs all genes $P < 2.2\text{E-}16$, upregulated genes vs all genes $P = 0.084$, downregulated genes vs upregulated genes $P < 2.2\text{E-}16$; *Setd1a/b* KD vs Control: downregulated genes vs all genes $P < 2.2\text{E-}16$, upregulated genes vs all genes $P = 0.11$, downregulated genes vs upregulated genes $P < 2.2\text{E-}16$). (H) A schematic outline the methodology for microinjection into MII oocytes, detailing the introduction of *Setd1a/b* siRNA to achieve knockdown (*Setd1a/b* KD), the co-injection of *Setd1a/b* siRNA with human WT *Setd1a/b* mRNA (*Setd1a/b* KD + hWT mRNA), and the combination of *Setd1a/b* siRNA with human MUT *Setd1a/b* mRNA (*Setd1a/b* KD + hMUT mRNA). (I) Images of early embryos from Control, *Setd1a/b* KD, *Setd1a/b* KD + *Setd1a/b* hWT mRNA, and *Setd1a/b* KD + *Setd1a/b* hMUT mRNA treatment, displaying a typical example from three independent experiments. Scale bar, 50 μm. (J) Line plot showing the development rate of embryos from Control, *Setd1a/b* KD, *Setd1a/b* KD + *Setd1a/b* hWT mRNA, and *Setd1a/b* KD + *Setd1a/b* hMUT mRNA groups. Error bars represent mean ± SD from three biological replicates. **$P < 0.01$; ***$P < 0.001$; two-sided unpaired Student's *t*-test (Control vs *Setd1a/b* KD $P = 0.00077$; *Setd1a/b* KD + *Setd1a/b* hWT mRNA vs *Setd1a/b* KD + *Setd1a/b* hMUT mRNA $P = 0.0014$). Source data are available online for this figure.

knockouts of *Setd1a* and *Setd1b* result in lethality at the E7.5 and E11.5 stages of mouse embryogenesis, respectively (Bledau et al, 2014). We further discovered that knockdown of *Setd1a* leads to compensatory expression of *Setd1b*, and simultaneous knockdown of both *Setd1a* and *Setd1b* significantly reduces the rate of blastocyst formation, a phenotype consistent with the results observed in *Wdr82* knockdown (Bi et al, 2011). Moreover, the knockdown of *Setd1a/b* leads to widespread downregulation of gene expression during the morula and blastocyst stages, including key lineage genes such as *Nanog*, *Pou5f1*, *Tead4*, and *Cdx2*, highlighting the critical role of SETD1A/B in the establishment of pluripotency. Previous research has already underscored the importance of the SETD1A/B complex components, including CXXC1 and SETD1A, in influencing cell differentiation, highlighting the methyltransferase activity of SETD1A as essential for the precise regulation of gene expression during the differentiation of ESCs (Fang et al, 2016; Lin et al, 2019; Sze et al, 2017). Despite these insights affirming the extensive influence of SETD1A/B-mediated H3K4me3 on cell differentiation, emerging studies suggest that SETD1A/B might exert effects through mechanisms not solely reliant on its methyltransferase activity (Hoshii et al, 2018; Morgan and Shilatifard, 2020; Sze et al, 2017). Consequently, we further demonstrated that overexpression of WT human SETD1A/B successfully rescued the development of mouse *Setd1a/b* knockdown embryos, whereas catalytically inactive mutants did not, confirming that SETD1A/B's methyltransferase activity is essential for regulating the transition from totipotency to pluripotency during early embryonic development.

Notably, our findings reveal that the transition from totipotency to pluripotency in early embryos is intricately regulated by the differential engagement of MLL2 and SETD1A/B in depositing the H3K4me3. Specifically, in the Late2C embryos, MLL2-mediated H3K4me3 deposition encompasses two distinct mechanisms: (i) MLL2, harboring a CXXC motif, preferentially targets areas rich in CpG sequences to deposit H3K4me3, aligning with shared H3K4me3 peaks (Hu et al,

2017); (ii) Additionally, MLL2 is likely directed by ZGA-specific factors to CpG-sparse regions to establish Late2C-specific H3K4me3 peaks, a phenomenon that merits further exploration. In morulae, both shared and Morula-specific H3K4me3 peaks are deposited by SETD1A/B in a transcription-dependent manner. The reliance of H3K4me3 deposition on transcription in morulae can be attributed, on the one hand, to the recruitment of the SETD1A/B complex to actively transcribed genes by Pol II (Bae et al, 2020; Muntean et al, 2010), and on the other hand, to the rapid demethylation by KDM5s following transcriptional inhibition, suggesting that transcription stabilizes H3K4me3 levels during this stage. We propose a model in which MLL2-mediated H3K4me3 deposition and transcription do not interact in totipotent zygotes and two-cell embryos. However, from the four-cell stage through to the morula, as cells shift from totipotency to pluripotency, transcription both promotes SETD1A/B-mediated H3K4me3 deposition and prevents demethylation by KDM5s. The H3K4me3 deposited in this manner further enhances transcription, establishing a positive feedback loop. Such precise regulation of H3K4me3 deposition is pivotal for finely tuning the transitions in cell potency, ensuring orderly developmental progression and cell lineage determination during early embryo development.

## Methods

### Reagents and tools table

| Reagent/resource | Reference or source | Identifier or catalog number |
|---|---|---|
| **Experimental models** | | |
| Mouse: C57BL/6J | Produced by the Laboratory Animal Center of Huazhong Agricultural University | N/A |
| Mouse: DBA/2 | Produced by the Laboratory Animal Center of Huazhong Agricultural University | N/A |

| Reagent/resource | Reference or source | Identifier or catalog number |
|---|---|---|
| Mouse: B6D2F1 | Produced by the Laboratory Animal Center of Huazhong Agricultural University | N/A |
| **Recombinant DNA** | | |
| RN3P-KDM5B-FLAG | Addgene | Cat #86398 |
| pcDNA3.1-mus-Kdm5b-WT | This study | N/A |
| pcDNA3.1-mus-Kdm5b-H499A | This study | N/A |
| pcDNA3.1-hus-SETD1A-WT | Suzhou Institute Plasmid Resource Sharing Platform | Cat #SP-1569 |
| pcDNA3.1-hus-SETD1A-ΔS1621/S1622I | This study | N/A |
| pCMV-hus-SETD1B-WT | MiaoLing Biology | Cat #P49194 |
| pCMV-hus-SETD1B-ΔS1880/S1881I | This study | N/A |
| **Antibodies** | | |
| Anti-Histone H3 (tri methyl K4) | Abcam | Cat #ab8580 |
| MLL4 Polyclonal Antibody | Invitrogen | Cat #PA5-103371 |
| Anti-hSET1 | Abcam | Cat #ab70378 |
| SETD1B Polyclonal antibody | Proteintech | Cat #55005-1-AP |
| PLU-1 Antibody (7H3D7) | Santa | Cat #sc-517291 |
| Anti-Nanog | Abcam | Cat #ab80892 |
| Anti-CDX-2 | Biogenex | Cat #MU392A-5UC |
| Dylight 488 goat anti-rabbit IgG | Abbkine | Cat #A23220 |
| Dylight 549 goat anti-rabbit IgG | Abbkine | Cat #A23320 |
| Dylight 488 goat anti-mouse IgG | Abbkine | Cat #A23210 |
| RNA pol II antibody | Active Motif | Cat #61668 |
| Unconjugated Secondary Antibody for CUT&Tag | Vazyme | Cat #Ab207-01 |
| **Oligonucleotides and other sequence-based reagents** | | |
| qPCR primers | This study | Dataset EV1 |
| small interference RNA | This study | Dataset EV1 |
| **Chemicals, enzymes, and other reagents** | | |
| Pregnant Mare Serum Gonadotropin | Ningbo Second Hormone Factory | N/A |
| Human Chorionic Gonadotropin | Ningbo Second Hormone Factory | N/A |
| G1-Plus medium | Vitrolife | Cat #10132 |
| HTF medium | Merck | Cat #MR-070 |
| hyaluronidase | Sigma | Cat #H3506 |
| T7 RNAi Transcription Kit | Vazyme | Cat #TR102 |
| mMESSAGE mMACHINE T7 Ultra Kit | Invitrogen | Cat #AM1345 |
| Triptolide | Sigma | Cat #T3652 |
| CPI-455 | Selleck | Cat #S6389 |
| RNAprep Pure Micro Kit | TIANGEN | Cat #DP420 |

| Reagent/resource | Reference or source | Identifier or catalog number |
|---|---|---|
| HiScript II Q RT SuperMix for qPCR Kit with gDNA wiper | Vazyme | Cat #R223-01 |
| ChamQ Universal SYBR qPCR Master Mix | Vazyme | Cat #Q321-02 |
| Cell-Light EU Apollo 567 In Vitro Imaging Kit | RiboBio | Cat #C10316-1 |
| Triton X-100 | Sigma | Cat #93443 |
| Recombinant RNase inhibitor | Takara | Cat #2313 A |
| Deoxynucleotide (dNTP) Solution Mix | NEB | Cat #N0447S |
| ERCC RNA Spike-In Mix | Thermo Fisher Scientific | Cat #4456740 |
| SuperScript™ II Reverse Transcriptase | Invitrogen | Cat #18064014 |
| Betaine | Sigma | Cat #61962 |
| MgCl$_2$ | Sigma | Cat #M1028 |
| KAPA HiFi HotStart ReadyMix | Roche | Cat #KK2605 |
| VAHTS DNA Clean Beads | Vazyme | Cat #N411-01 |
| TruePrep DNA Library Prep Kit | Vazyme | Cat #TD502 |
| Dynabeads™ Protein G for Immunoprecipitation | Invitrogen | Cat #10003D |
| Phenol–chloroform–isoamyl alcohol mixture | Sigma | Cat #77617 |
| KAPA Hyper Prep Kit | Roche | Cat #KK8504 |
| Hyperactive In-Situ ChIP Library Prep Kit for Illumina | Vazyme | Cat #TD901 |
| High-Fidelity 2X PCR Master Mix | NEB | Cat #M0541S |
| TruePrep Index Kit V2 for Illumina | Vazyme | Cat #TD202 |
| **Software** | | |
| TrimGalore V0.6.6 | https://www.bioinformatics.babraham.ac.uk/projects/trim_galore/ | N/A |
| STAR V2.7.3a | https://github.com/alexdobin/STAR/ (Dobin et al, 2013) | N/A |
| featureCounts V1.6.2 | https://subread.sourceforge.net/ (Liao et al, 2014) | N/A |
| Homer V4.11 | http://homer.ucsd.edu/homeh/ (Heinz et al, 2010) | N/A |
| RUVSeq V1.6.2 | https://www.bioconductor.oro/packages/release/bioc/htht/RUVSeq.html (Risso et al, 2014) | N/A |
| DESeq2 V1.30.1 | https://bioconductor.org/packages/release/bioc/html/DESeq2.html (Love et al, 2014) | N/A |
| Bowtie2 V2.4.1 | https://sourceforge.net/projects/bowtie-bio/files/bowtie2/ (Langmead and Salzberg, 2012) | N/A |
| SAMtools V1.9 | http://www.htslib.org/ (Li et al, 2009) | N/A |
| Picard V2.23.9 | https://github.com/broadinstitute/picard | N/A |
| deepTools V3.5.0 | https://github.com/deeptools/deepTools (Ramirez et al, 2014) | N/A |

| Reagent/resource | Reference or source | Identifier or catalog number |
|---|---|---|
| bedtools V2.27.0 | https://bedtools.readthedocs.io/en/latest/index.html (Quinlan and Hall, 2010) | N/A |
| SRA Toolkit V2.9.6 | https://github.com/ncbi/sra-tools | N/A |
| Integrative Genomics Viewer V2.6.2 | http://software.broadinstitute.org/software/igv/ (Robinson et al, 2011) | N/A |
| MACS2 V2.2.7.1 | https://github.com/taoliu/MACS/ (Zhang et al, 2008) | N/A |
| SICER V1.0.2 | https://zanglab.github.io/SICER2/ (Zang et al, 2009) | N/A |
| ChIPseeker V1.26.2 | https://bioconductor.org/packages/release/bioc/html/ChIPseeker.html (Yu et al, 2015) | N/A |
| clusterProfiler V3.18.1 | https://www.bioconductor.org/packages/release/bioc/html/clusterProfiler.html (Yu et al, 2012) | N/A |
| GraphPad Prism V8.0.1 | https://www.graphpad.com/scientificsoftware/prism | N/A |
| R V4.0.2 | https://www.r-project.org/ | N/A |
| ImageJ | https://imagej.net/software/imagej/ | N/A |
| Other | | |
| Illumina NovaSeq 6000 | Illumina | N/A |
| PiezoXpert micromanipulator | Eppendorf | N/A |
| Confocal microscope | Zeiss | LSM 800 |

No statistical methods were used to predetermine the sample size. The experiments were not randomized, and the investigators were not blinded to allocation during outcome assessment.

## Animals and collection of mouse embryos

Specific-pathogen-free mice were housed in Huazhong Agricultural University's animal facility in Wuhan, China. All experimental protocols adhered to the guidelines of Huazhong Agriculture University's Animal Care and Use Committee (Approval No: HZAUMO-2023-0106). For early embryo collection, 6- to 8-week-old B6D2F1 (C57BL/6J × DBA/2) female mice underwent superovulation by injection with pregnant mare serum gonadotropin (PMSG, 10 IU) (Ningbo Second Hormone Factory, China), followed by injection with human chorionic gonadotropin (hCG, 10 IU) (Ningbo Second Hormone Factory, China) 48 h later. MII oocytes were collected from the oviducts of non-mated females, while zygotes were obtained from the female mice were mated with B6D2F1 males. Embryos were cultured in G1-Plus medium (Vitrolife, 10132) at 37 °C in a 5% $CO_2$ atmosphere, and collected at specific time points: Early2C embryos at 30 h, Late2C embryos at 44 h, morulae at 78 h, and blastocysts at 96 h post-hCG injection.

For in vitro fertilization (IVF), sperm from the cauda epididymis of B6D2F1 males was prepared in HTF medium (Merck, MR-070) and incubated for 60 min at 37 °C in 5% $CO_2$ to enable sperm capacitation. MII oocytes were then stripped of cumulus cells via hyaluronidase (Sigma, H3506) treatment, and subjected to zona pellucida perforation using a PiezoXpert micromanipulator (Eppendorf). Subsequently, these oocytes were incubated in a pre-prepared B6D2F1 sperm suspension within HTF medium for 4 h, washed, and cultured in a G1-Plus medium. After IVF, embryos were harvested at predetermined times: Early2C embryos

at 18 h, Late2C embryos at 32 h, morulae at 66 h, and blastocysts at 84 h post-fertilization.

## In vitro transcription and microinjection

Target-specific interference sequences for mouse *Mll2*, *Setd1a*, *Kdm5a*, and *Kdm5b* were designed using the DSIR siRNA design platform, with three pairs per target. The *Setd1b* interference sequences were derived from previous studies (Redd et al, 2017). Target-specific DNA oligonucleotides were annealed and subsequently synthesized into siRNA using the T7 RNAi Transcription Kit (Vazyme, TR102). In the knockdown study, siRNAs were utilized at a working concentration of 25 μM, with non-targeting siRNA as the control. mRNA was synthesized following the protocol provided by the mMESSAGE mMACHINE T7 Ultra Kit (Invitrogen, AM1345). The WT *Kdm5b* cDNA was acquired from Addgene (86398) (Zhang et al, 2016), the WT *SETD1A* cDNA was sourced from the Suzhou Institute Plasmid Resource Sharing Platform (SP-1569), and the WT *SETD1B* cDNA was obtained from MiaoLing Biology (P49194). To generate catalytically inactive variants of KDM5B, SETD1A, and SETD1B, we introduced the mutations H499A, ΔS1621/S1622I, and ΔS1880/S1881I, respectively, using site-directed mutagenesis. The injection concentrations were set at 800 ng/μL for *Kdm5b* and *SETD1A* mRNA and 200 ng/μL for *SETD1B* mRNA. Approximately 10 pL of siRNAs or mRNA were microinjected into zygotes or MII oocytes using a PiezoXpert micromanipulator (Eppendorf), followed by culture in G1-Plus medium at 37 °C in a 5% $CO_2$ environment. Details of the siRNA sequences are listed in Dataset EV1.

## Treatment with triptolide and CPI-455

For embryonic treatments with inhibitor, Triptolide (Sigma, T3652) was solubilized in dimethyl sulfoxide (DMSO) to prepare a 10 mM stock solution, which was subsequently diluted in G1-Plus medium to achieve final concentrations of 0.01 μM, 0.1 μM, or 1 μM. Similarly, CPI-455 (Selleck, S6389) was prepared as a 50 mM stock solution in DMSO and diluted with G1-Plus medium to final concentrations of 10 μM, 25 μM, or 50 μM. A 0.1% DMSO solution served as the control for DMSO exposure.

## Reverse transcription and quantitative PCR (RT-qPCR) analysis

Total RNA extraction from 30 embryos using the RNAprep Pure Micro Kit (TIANGEN, DP420). Subsequent synthesis of complementary DNAs (cDNAs) was performed with the HiScript II Q RT SuperMix for qPCR Kit with gDNA wiper (Vazyme, R223-01), followed by quantification via ChamQ Universal SYBR qPCR Master Mix (Vazyme, Q321-02) on a CFX96 Real-Time PCR Detection System (Bio-Rad). Control embryo results were set as a baseline of 1 and normalized against the internal control gene *H2afz*. Data are reported as fold change (FC) = $2^{-\Delta\Delta Ct}$, presented as mean ± standard deviation (SD). Primer sequences can be found in Dataset EV1.

## Immunofluorescence

Embryos were fixed in 4% paraformaldehyde (PFA) for 30 min at room temperature, followed by washing with 0.05% polyvinyl alcohol

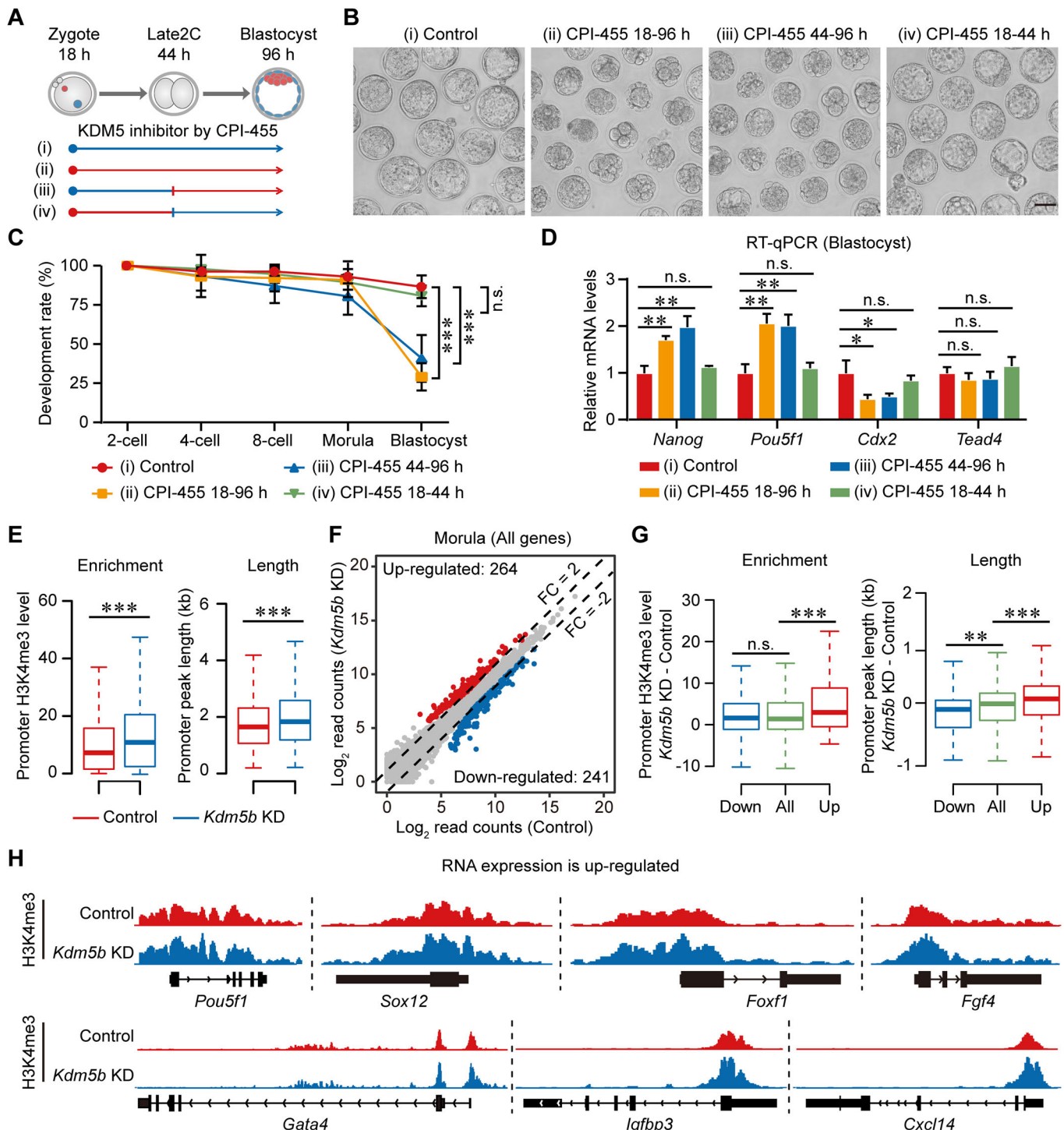

**A** Zygote 18 h — Late2C 44 h — Blastocyst 96 h

KDM5 inhibitor by CPI-455
(i), (ii), (iii), (iv)

**B** (i) Control (ii) CPI-455 18-96 h (iii) CPI-455 44-96 h (iv) CPI-455 18-44 h

**C** Development rate (%) vs 2-cell, 4-cell, 8-cell, Morula, Blastocyst
(i) Control
(ii) CPI-455 18-96 h
(iii) CPI-455 44-96 h
(iv) CPI-455 18-44 h

**D** RT-qPCR (Blastocyst) — Relative mRNA levels — Nanog, Pou5f1, Cdx2, Tead4
(i) Control
(ii) CPI-455 18-96 h
(iii) CPI-455 44-96 h
(iv) CPI-455 18-44 h

**E** Enrichment — Promoter H3K4me3 level; Length — Promoter peak length (kb)
Control / Kdm5b KD

**F** Morula (All genes)
Up-regulated: 264
Down-regulated: 241
$FC = 2$, $FC = -2$
$Log_2$ read counts (Kdm5b KD) vs $Log_2$ read counts (Control)

**G** Enrichment — Promoter H3K4me3 level Kdm5b KD − Control; Length — Promoter peak length (kb) Kdm5b KD − Control
Down, All, Up

**H** RNA expression is up-regulated
H3K4me3 Control / Kdm5b KD
Pou5f1, Sox12, Foxf1, Fgf4
Gata4, Igfbp3, Cxcl14

in PBS. The samples were then permeabilized with 0.5% Triton X-100 in PBS for 30 min. Blocking was performed using 5% bovine serum albumin (BSA) in PBS for 2 h before overnight incubation at 4 °C with primary antibodies (H3K4me3 (Abcam, ab8580), MLL2 (Invitrogen, PA5-103371), SETD1A (Abcam, ab70378), SETD1B (Proteintech, 55005-1-AP), KDM5B (Santa, sc-517291), NANOG (Abcam, ab80892), CDX2 (Biogenex, MU392A-5UC) at a 1:200 dilution in 5% BSA. Following PVA (0.05% PVA in PBS) washes, samples were

exposed to secondary antibodies in 5% BSA for 1 h, employing Dylight 488/549 goat anti-rabbit IgG (Abbkine, A23220/A23320) and Dylight 488 goat anti-mouse IgG (Abbkine, A23210; 1:500 dilution) for immunostaining. After three subsequent washes, specimens were mounted on slides using an anti-fade mounting medium with DAPI (Beyotime, P0131).

To assess RNA synthesis, Early2C embryos, Late2C embryos, and morulae were incubated in a G1-Plus medium added with 500 μM

**Figure 6. Essential role of H3K4me3 regulation during post-ZGA for the first lineage segregation.**

(A) Schematic illustrates CPI-455 treatment protocols in early embryonic development stages: (i) DMSO treatment from zygote to blastocyst, (ii) CPI-455 treatment from zygote to blastocyst, (iii) CPI-455 treatment from Late2C to blastocyst, and (iv) CPI-455 treatment from zygote to Late2C. Periods of CPI-455 treatment are marked in red. (B) Representative images of blastocysts from four treatment groups. One representative image from four independent experiments is shown. Scale bar, 50 μm. (C) Line plots showing the development rates of embryos from the four treatment groups at specific time points. Error bars represent mean ± SD from four biological replicates. n.s., not significant; ***$P < 0.001$; two-sided unpaired Student's $t$-test (CPI-455 18–96 h vs Control $P = 0.000046$, CPI-455 44–96 h vs Control $P = 0.00013$, CPI-455 18–44 h vs Control $P = 0.28$). (D) Bar graphs display the expression levels of *Nanog*, *Pou5f1*, *Cdx2*, and *Tead4* in blastocysts from each treatment groups, as detected by RT-qPCR. Error bars represent mean ± SD from three biological replicates. n.s., no significance; *$P < 0.05$; **$P < 0.01$; two-sided unpaired Student's $t$-test (*Nanog*: CPI-455 18–96 h vs Control $P = 0.0024$, CPI-455 44–96 h vs Control $P = 0.0038$, CPI-455 18–44 h vs Control $P = 0.23$; *Pou5f1*: CPI-455 18–96 h vs Control $P = 0.0030$, CPI-455 44–96 h vs Control $P = 0.0048$, CPI-455 18–44 h vs Control $P = 0.46$; *Cdx2*: CPI-455 18–96 h vs Control $P = 0.030$, CPI-455 44–96 h vs Control $P = 0.039$, CPI-455 18–44 h vs Control $P = 0.41$; *Tead4*: CPI-455 18–96 h vs Control $P = 0.29$, CPI-455 44–96 h vs Control $P = 0.37$, CPI-455 18–44 h vs Control $P = 0.32$). (E) Boxplot showing the H3K4me3 enrichment and peak length at the promoter regions ($n = 22,470$; TSS ± 2 kb) in Control and *Kdm5b* KD morulae. Boxplot was used to display data distribution, with the median as the central line, the box showing the IQR from the 25th to 75th percentile, and whiskers extending to data points within 1.5 times the IQR. ***$P < 0.001$; two-sided Wilcoxon–Mann–Whitney test (Enrichment: $P < 2.2E-16$; Length: $P < 2.2E-16$). (F) Scatter plots comparing the gene expression levels between Control and *Kdm5b* KD morulae, highlighting genes upregulated (red) and downregulated in *Kdm5b* KD embryos (blue). (G) Box plots showing the H3K4me3 enrichment levels at the promoter regions (TSS ± 2 kb) of downregulated genes ($n = 241$), upregulated genes ($n = 264$), and all genes ($n = 22,470$) when comparing Control with *Kdm5b* KD morulae. Boxplot was used to display data distribution, with the median as the central line, the box showing the IQR from the 25th to 75th percentile, and whiskers extending to data points within 1.5 times the IQR. n.s., not significant; **$P < 0.01$; ***$P < 0.001$; two-sided Wilcoxon–Mann–Whitney test (Enrichment: downregulated genes vs all genes $P = 0.91$, upregulated genes vs all genes $P = 6.7E-06$; Length: downregulated genes vs all genes $P = 0.0067$, upregulated genes vs all genes $P = 9.3E-05$). (H) Genome browser views showing the H3K4me3 enrichments for representative upregulated genes in Control and *Kdm5b* KD morulae. Source data are available online for this figure.

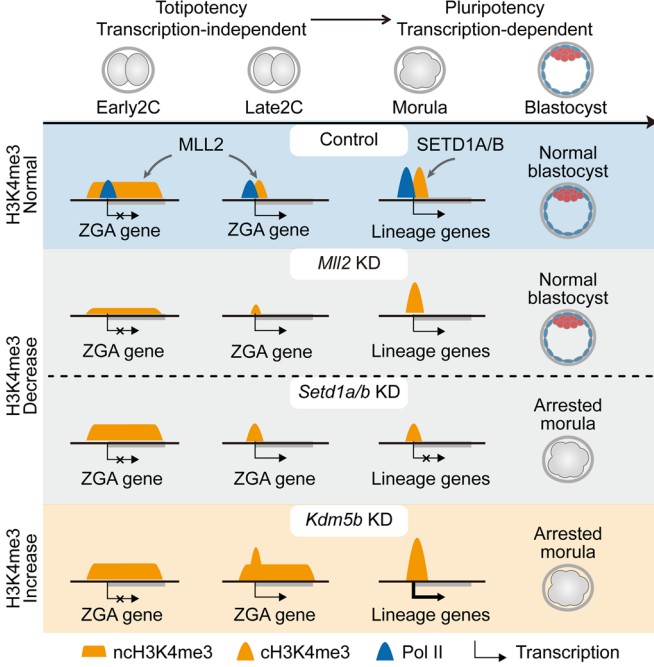

**Figure 7. Models of the dynamic establishment and functions of H3K4me3 in mouse early embryos.**

During the transition from totipotency to pluripotency in early embryos, H3K4me3 deposition shifts from MLL2-mediated, transcription-independent mechanisms to transcription-dependent deposition by SETD1A/B. Disruption of MLL2-mediated H3K4me3 deposition does not impact ZGA gene transcription and pre-implantation development. However, disruption of SETD1A/B-mediated H3K4me3 deposition is critical for the expression of key genes involved in the first cell fate decision and pre-implantation development.

5-ethynyl uridine (EU) for 2 h at 37 °C prior to fixation. EU integration was visualized using the Cell-Light EU Apollo 567 In Vitro Imaging Kit (RiboBio, C10316-1), following the manufacturer's protocol.

Fluorescence observations were made with a confocal microscope (Zeiss, LSM 800), and fluorescence intensity was analyzed using Fiji

software. The signal intensity within nuclei and cytoplasm of embryos was assessed, with cytoplasmic signal deducted from nuclear signal to adjust for background. The number of cells in each embryo was ascertained by counting DAPI-stained nuclei.

## Smart-seq2

The RNA-seq libraries were prepared according to the Smart-seq2 protocol with certain modifications (Picelli et al, 2014). Briefly, 10 embryos at the Early2C, Late2C, morula, and blastocyst stages were collected. The zona pellucida was removed using 0.5% pronase E, followed by three washes in 0.05% polyvinyl alcohol. Embryos were then placed into 4 μL of lysis buffer, which included 2 μL of 0.2% Triton X-100 (Sigma, 93443), 4 U of RNase inhibitor (Takara, 2313 A), 1 μL of 100 μM oligo-dT primer (5'-AAGCAGTGGTAT-CAACGCAGAGTACT30VN-3'), 1 μL of 1 mM dNTP (NEB, N0447S), and 0.05 μL of 1:1000 diluted ERCC spike-in (Thermo Fisher Scientific, 4456740). After heating at 72 °C for 3 min, the lysis mixture was supplemented with 5.7 μL of reverse transcription mix, containing 100 U SuperScript II reverse transcriptase (Invitrogen, 18064014), 1× Superscript II first-strand buffer, $5 \times 10^{-3}$ M DTT, 1 M betaine (Sigma, 61962), $6 \times 10^{-3}$ M MgCl$_2$ (Sigma, M1028), $1 \times 10^{-6}$ M TSO (5'-AAGCAGTGGTATCAACG-CAGAGT-3'), and 10 U RNase inhibitor, and incubated at 42 °C for 90 min for cDNA synthesis. This was followed by 16–18 cycles of preamplification using KAPA HiFi HotStart ReadyMix (Roche, KK2605) and IS PCR primers (5'-AAGCAGTGGTATCAACGCA-GAGT-3'), and purification with VAHTS DNA Clean Beads (Vazyme, N411-01). For library construction, 1 ng of the amplified cDNA was fragmented and processed using the TruePrep DNA Library Prep Kit (Vazyme, TD502), following the manufacturer's guidelines. Sequencing was conducted on the Illumina NovaSeq 6000 system, generating paired-end 150-bp reads.

## ULI-NChIP-seq

ULI-NChIP-seq was performed in accordance with a modified version of a previously established protocol (Liu et al, 2016). For each

immunoprecipitation, 200–500 embryonic cells were collected. The zona pellucida was removed using 0.5% pronase E, followed by triple rinsing in 0.05% polyvinyl alcohol. The embryos were then incubated in a $Ca^{2+}$-free CZB medium at 37 °C for 5 min, with subsequent gentle pipetting employed to dislodge polar bodies. Embryos were mixed with 20 μL Nuclear Isolation Buffer and 30 μL MNase Master Mix (10 U/mL) for chromatin digestion 7 min at 25 °C. Digestion was halted by adding 5.5 μL of 0.1 M EDTA and 5.5 μL of a 1% Triton X-100 plus 1% deoxycholate solution. Antibody-coated bead complexes, prepared by incubating 11 μL Dynabeads Protein G (Invitrogen, 10003D) with 1 μg of Pol II antibodies (Active Motif, 61668) in Complete Immunoprecipitation Buffer for 6 h at 4 °C, were then added to the samples. Following overnight incubation at 4 °C, the samples underwent two washes with Low Salt and High Salt Wash Buffers, respectively. Chromatin was eluted in 100 μL ChIP Elution Buffer at 65 °C for 2 h. DNA, with a 2 μL spike-in, was extracted using phenol–chloroform–isoamyl alcohol (25:24:1) (Sigma, 77617) and processed for library preparation with the KAPA HyperPrep Kit (Roche, KK8504) as per the manufacturer's protocol. Sequencing was conducted using paired-end 150-bp reads on the Illumina NovaSeq 6000 platform.

## CUT&Tag

CUT&Tag was performed according to a method previously described (Kaya-Okur et al, 2019), using the Hyperactive In-Situ ChIP Library Prep Kit for Illumina (Vazyme, TD901). In brief, 200–500 cells were incubated with 10 μL balanced concanavalin A-coated magnetic beads in a 0.2 mL low-binding tube. To this, 50 μL of antibody buffer containing 1 μg of the H3K4me3 antibody (Abcam, ab8580) was added, and the mixture was incubated overnight at 4 °C. After washing three with dig-wash buffer, 50 μL dig-wash buffer containing 0.5 μg of a secondary antibody (goat anti-rabbit IgG, Vazyme, Ab206-10-AA) was added and incubated at room temperature for 1 h. After another three washes with 200 μL dig-wash buffer, 0.58 μL of pG-Tn5 and 100 μL of dig-300 buffer were added. The samples were incubated at room temperature for 1 h, followed by two washes with 200 μL dig-wash buffer. Tagmentation was carried out by adding 300 μL of tagmentation buffer and incubating at 37 °C for 1 h, before stopping the reaction with 10 μL of 0.5 M EDTA, 3 μL of 10% SDS, and 2.5 μL of 20 mg/mL Proteinase K. DNA, enriched with 2 μL of spike-in DNA, was extracted using 100 μL of phenol–chloroform–isoamyl alcohol (25:24:1) (Sigma, 77617). Amplification was then conducted with 0.5 μL of each primer (primer 1: 5′-TCGTCGGCAGCGTCAGATGTGTATAAGAGA-CAG-3′, primer 2: 5′-GTCTCGTGGGCTCGGAGATGTGTATAA-GAGACAG-3′) and 25 μL of high-fidelity DNA polymerase (NEB, M0541S). Library construction utilized 5 μL of indexes (Vazyme, TD202) and 25 μL of High-Fidelity DNA Polymerase. Sequencing was executed on the Illumina NovaSeq 6000 system, producing paired-end 150-bp reads.

## Preparation of ULI-NChIP-seq and CUT&Tag spike-in DNA

*Drosophila* genomic DNA was employed as spike-in control, according to a protocol previously detailed (Skene and Henikoff, 2017). Briefly, 5 ng of *Drosophila* genomic DNA was fragmented

with Tn5 transposase (Vazyme, TD502) and purified through phenol–chloroform extraction, followed by ethanol precipitation. The purified DNA was then resuspended in 1 mL of $H_2O$. For the ULI-NChIP-seq and CUT&Tag procedures, 2 μL of this solution, diluted 1:1000, was used as the spike-in material.

## RNA-seq data analysis

Adapters and low-quality reads were removed using TrimGalore (version 0.6.6). Trimmed reads were mapped to mm10 mouse genome (including ERCC spike-in sequences) using STAR (version 2.7.3a). The number of reads uniquely mapped to mm10 gene annotations and ERCC transcripts was determined using feature-Counts (version 1.6.2). The function analyzeRepeats.pl from Homer (version 4.11) software was used to get raw counts for repeats. Raw read counts were ERCC spike-in-normalized using RUVSeq (version 1.6.2), and differential expression analysis was then performed using DESeq2 (version 1.30.1). Genes or repeats with a Benjamini and Hochberg-adjusted $P$ value ≤0.05 and an absolute LFC value ($\log_2[FC]$) >1 were considered as differentially expressed genes or repeats. Normalized read counts were analyzed using the plotPCA function in the DESeq2 (version 1.30.1) package for principal component analysis (PCA), and Spearman's rank coefficients were computed using the "cor" function.

## Spike-in normalization for ChIP-seq data analysis

Adapters and low-quality reads were removed using TrimGalore (version 0.6.6). Trimmed reads were individually aligned to mm10 mouse genome or dm6 *Drosophila* genome using Bowtie2 (version 2.4.1) with the options "--no-mixed" and "--no-discordant". Reads with low quality (MAPQ <30) were filtered out using SAMtools (version 1.9), and PCR duplicates were removed using Picard (version 2.23.9). Spike-in dm6 reads were quantified using SAMtools (version 1.9), and the scale factor α = 1e6/dm6_count was calculated, where dm6_count indicates the total spike-in reads in a sample. To standardize the control group to 1, we adjusted each sample's normalization factor to the control group's mean normalization factor. Normalized reads per kilobase of bin per million mapped reads (RPKM) bigwig files were generated using the bamCoverage function from deepTools (version 3.5.0), applying the previously calculated scale factors. Pearson correlations were calculated using log2-transformed RRPM values in 5-kb bins across the genome. Heatmaps and metagene plots were generated using the computeMatrix function followed by the plotHeatmap and plotProfile functions of deepTools (version 3.5.0). The CpG density was determined by calculating the total number of CpG sites within 2-kb upstream and downstream of the center of each H3K4me3 peak using bedtools (version 2.27.0).

## ChIP-seq data analysis

Downloaded published H3K4me3, Pol II, H3K27ac, H3K9ac, and H2A.Z ChIP-seq raw data, and converted them to FASTQ format using the SRA Toolkit (version 2.9.6). All reads were mapped to the mm10 mouse genome using Bowtie2 (version 2.4.1) with the options "--no-mixed" and "--no-discordant". The SAMtools (version 1.9) was used to remove low-quality reads (MAPQ <30) and Picard (version 2.23.9) was used to remove PCR duplicates.

RPKM bigwig files were generated by bamCoverage subcommand in deepTools (version 3.5.0), and the tracks were visualized with Integrative Genomics Viewer (IGV, version 2.6.2).

### Identification of Late2C-specific, shared and morula-specific H3K4me3 peaks

Canonical H3K4me3 peaks for the Late2C and morula stages were called using MACS2 (version 2.2.7.1). Subsequently, the H3K4me3 peaks from both Late2C and morula stages were combined, and RPKM values for these regions were calculated. Stage-specific peaks were chosen based on the following criteria: an RPKM >1.5 at the H3K4me3 peak for the respective stage and an RPKM <0.5 at the other stage. Peaks with RPKM >1.5 at both stages were designated as shared peaks.

### Identification of ncH3K4me3 and cH3K4me3 peaks in MII oocytes and Late2C embryos

The SICER algorithm (version 1.0.2) was employed, employing a 500 bp window and a 5000 bp gap, to identify broad H3K4me3 peaks in MII oocytes, categorizing those exceeding 5 kb in width as ncH3K4me3 peaks. MACS2 (v2.2.7.1) was deployed for the delineation of narrow H3K4me3 peaks in Late2C embryos, designated as cH3K4me3 peaks.

Transitioning from MII oocytes to Late2C embryos, genes are classified into C1, those losing ncH3K4me3 peaks, and C2, those gaining cH3K4me3 peaks, based on their promoter overlap (TSS ± 2 kb) with H3K4me3 peaks.

### Identification of ZGA, ICM-specific, TE-specific, and expressed genes

We used a predefined ZGA gene list obtained from a previous study (Wang et al, 2022). The RNA-seq (Wang et al, 2018) data for early embryo development was reanalyzed, considering genes with an FC >2 and FPKM >5 between ICM and TE as ICM- and TE-specific genes.

### Peak annotation and gene ontology analysis

Annotate H3K4me3 peaks to the nearest genes using ChIPseeker (version 1.26.2). Gene ontology analysis was performed using the enrichGO function of the clusterProfiler (version 3.18.1) package.

### Statistics analysis

ULI-NChIP-seq was repeated once. Except for the ULI-NChIP-seq, all experiments were repeated at least twice times. $P$ values were determined by a two-sided Wilcoxon–Mann–Whitney test in R or a two-sided unpaired Student's $t$-test using GraphPad Prism (version 8.0.1). Significant differences were shown with *, **, and *** for indicating $P < 0.05$, 0.01, and 0.001, respectively. n.s. denotes no significance.

## Data availability

All data have been deposited to the Genome Sequence Archive database with the accession number CRA015496.

The source data of this paper are collected in the following database record: biostudies:S-SCDT-10_1038-S44318-024-00329-5.

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

## Acknowledgements

We thank Prof. Yawei Gao (Tongji University, Shanghai, China) for providing guidance. The computations in this paper were run on the bioinformatics computing platform of the National Key Laboratory of Crop Genetic Improvement, Huazhong Agricultural University. This work was supported by the National Natural Science Foundation of China (32425051 and 32372883), the National Key R&D Program of China (2022YFD1302200), the Fundamental Research Funds for the Central Universities (2662023DKPY001 and 2662021DKQD004).

## Author contributions

**Jingjing Zhang**: Conceptualization; Data curation; Formal analysis; Validation; Investigation; Visualization; Writing—original draft; Writing—review and editing. **Qiaoran Sun**: Formal analysis; Investigation; Writing—original draft; Writing—review and editing. **Liang Liu**: Formal analysis; Investigation. **Shichun Yang**: Formal analysis; Investigation. **Xia Zhang**: Formal analysis; Investigation. **Yiliang Miao**: Conceptualization; Resources; Data curation; Supervision; Funding acquisition; Methodology; Project administration; Writing—review and editing. **Xin Liu**: Conceptualization; Resources; Data curation; Supervision; Funding acquisition; Project administration; Writing—review and editing.

Source data underlying figure panels in this paper may have individual authorship assigned. Where available, figure panel/source data authorship is listed in the following database record: biostudies:S-SCDT-10_1038-S44318-024-00329-5.

## Disclosure and competing interests statement

The authors declare no competing interests.

