## [Review Process File · The EMBO Journal]

Histone methyltransferases MLL2 and SETD1A/B play distinct roles in H3K4me3 deposition during the transition from totipotency to pluripotency

Xin Liu, Jingjing Zhang, Qiaoran Sun, Liang Liu, Shichun Yang, Xia Zhang, and Yiliang Miao

Corresponding authors: Xin Liu (victorlau@mail.hzau.edu.cn) , Yiliang Miao (miaoyl@mail.hzau.edu.cn), Xin Liu (victorlau@mail.hzau.edu.cn)

Review Timeline:

Submission Date:	12th Apr 24
Editorial Decision:	24th May 24
Revision Received:	17th Sep 24
Editorial Decision:	8th Nov 24
Revision Received:	11th Nov 24
Accepted:	19th Nov 24

Editor: Ieva Gailite

Transaction Report:

Dear Dr. Miao,

Thank you for submitting your manuscript for consideration by the EMBO Journal. We have now received a full set of reviewer reports, which are included below for your information.

As you will see from the reports, reviewers #1 and #2 find the study per se of interest, while also pointing out several important controls and further experiments that are required to ensure the conclusiveness of the study. Furthermore, reviewers #2 and #3 indicate that previous literature and findings need to be appropriately discussed in the manuscript, and mention several instances where terminology needs to be more clearly defined.

Based on the interest expressed in the reports of reviewers #1 and #2, I would like to invite you to address the issues raised by all referees in a revised manuscript. Please note that the final decision will depend on the reviewers' assessment of the revised version, and in particular the inclusion of appropriate controls will be crucial for acceptance here. I think it would be useful to discuss the revision in more detail via email or phone/videoconferencing - please let me know which option you prefer.

We generally allow three months as standard revision time. As a matter of policy, competing manuscripts published during this period will not negatively impact on our assessment of the conceptual advance presented by your study. However, please contact me as soon as possible upon publication of any related work to discuss the appropriate course of action. Should you foresee a problem in meeting this three-month deadline, please contact us to arrange an extension.

When preparing your letter of response to the referees' comments, please bear in mind that this will form part of the Review Process File and will therefore be available online to the community. For more details on our Transparent Editorial Process, please visit our website: <https://www.embopress.org/page/journal/14602075/authorguide#transparentprocess>. Please also see the attached instructions for further guidelines on preparation of the revised manuscript.

Please feel free to contact me if you have any further questions regarding the revision. Thank you for the opportunity to consider your work for publication. I look forward to discussing your revision.

Yours sincerely,

Ieva Gailite

- a point-by-point response to the referees' comments, with a detailed description of the changes made (as a word file).
- a word file of the manuscript text.
- individual production quality figure files (one file per figure)

- a complete author checklist, which you can download from our author guidelines (<https://www.embopress.org/page/journal/14602075/authorguide>).

- Expanded View files (replacing Supplementary Information)

We realize that it is difficult to revise to a specific deadline. In the interest of protecting the conceptual advance provided by the work, we recommend a revision within 3 months (22nd Aug 2024). Please discuss the revision progress ahead of this time with the editor if you require more time to complete the revision.

Referee #1:

The authors focused on the biological significance of non-canonical (nc) and canonical (c) H3K4me3 distribution as well as its transition. Mll2 dictated cH3K4me3 distribution in a transcription-independent manner. In contrast, Setd1a/b maintained cH3K4me3 distribution in a transcription-dependent manner. Interestingly, Pol II inhibition by Triptolide for 2h was sufficient to reduce cH3K4me3 enrichment in morula embryos. Additional analyses revealed that the reduction of nc/cH3K4me3 enrichment by Mll2 KD did not affect the transcriptome of 2-cell embryos, which was also the case in Kdm5b-overexpressed embryos. Thus, global reduction of H3K4me3 did not affect ZGA. Next, the authors instead performed Kdm5b KD and observed the enhancement of ncH3K4me3 and cH3K4me3 enrichment in late 2-cell embryos. However, there was no significant correlation between changes in H3K4me3 and the alteration of gene expression. Thus, reduced as well as enhanced H3K4me3 enrichment in 2-cell embryos did not affect ZGA. In contrast, Setd1a/b KD impaired development to the blastocyst stage and affected lineage specification. Notably, decrease in Setd1a/b-dependent H3K4me3 was linked to the down-regulation of gene expression including Pou5f1, Nanog, and Cdx2. Kdm5b KD also impaired development to the blastocyst stage. However, in this case, Nanog-positive cells increased. Inhibitor assays showed that Kdm5b activity is required after the late 2-cell stage rather than the earlier period.

This paper enthusiastically addressed the biological significance of H3K4me3 in early mouse embryos. Overall, the authors' data provided are convincing, and the provided results support the authors' conclusion. Thus I support the publication of this paper in EMBO J, however, I suggest the authors to address the following points before the publication.

1. Fig. 1B,C,2D - Show changes in H3K4me3 enrichment at genome-wide bins as shown in Fig S2B, as it is unclear whether Mll2 KD or Setd1a/b KD causes only the global reduction of H3K4me3. I assume that H3K4me3 at some genomic regions are resistant to these KD. To address this point, please show which regions are preferentially affected or unaffected.
2. Fig. 2C,D - It is intriguing that H3K4me3 disappears within 2h treatment of Trp in morulae. However, further analysis was not performed in the current manuscript. I strongly suggest the authors to address the underlying mechanism of this phenomenon with additional experiments. Although Pol II-mediated recruitment of Setd1a/b may be operative (as described in Discussion), this discussion only would not be enough to explain this observation. For example, the authors can address the involvement of KDMs in the reduction of H3K4me3 caused by Trp.
3. Fig. 3 - The authors' data showing H3K4me3 is dispensable for major ZGA is convincing. However, it is unclear whether Kdm5b OE promotes the non-canonical-to-canonical transition precociously or just globally reduce the enrichment of H3K4me3. This investigation would be important to interpret the data.
4. Fig. 4F - Please show the commonality of DEGs between KD and inhibitor-treated samples by venn diagrams.
5. Fig. S10 - Effect of Kdm5b KD on H3K4me3 and development is highly convincing as the authors provide the results of rescue experiments. I strongly suggest the authors to perform H3K4me3 CUT&TAG and RNA-seq using Kdm5b KD morula samples. This will reveal how the absence of Kdm5b alters H3K4me3 distribution and whether enhanced H3K4me3 enrichment directly affects gene expression. I believe that this is a bit tough experiment for a revision but is required to meet the criteria of EMBO J.

6. Fig. 6E - Please include the illustration summarizing the results from Kdm5b OE and KD experiments.

Referee #2:

This manuscript comprises a series of knock-downs and over-expression studies to examine the contributions of the H3K4 methyltransferases MLL2, and SETD1A and SETD1B in mouse preimplantation embryos, and how they contribute to the changes in deposition of H3K4me3, including the transition for 'non-canonical' (i.e., with pervasive broad, intergenic domains) to 'canonical' (at promoters and CpG islands), and the consequential impact on fidelity of zygotic genome activation. Prior work has shown that MLL2 is an essential factor for oocyte development and competence (Ref. 33), and is responsible for the deposition of 'non-canonical' H3Kme3 independent of transcription in oocytes (Ref. 10). Separate zygotic genetic knock-outs of *Setd1a* and *Setd1b* have shown that both knock-outs are compatible with preimplantation development to blastocysts, but *Setd1a* is essential for further development of ICM, whereas *Setd1b* becomes essential only later in development (Ref. 39). Notably, combined oocyte and zygotic knock-out of *Setd1a* was reported not to lead to a more severe development phenotype, suggesting that maternally provided SETD1A is dispensable in the preimplantation embryo. In comparison, the original genetic knock-out of *Cxxc1/Cfp1*, which is required by both SETD1A and SETD1B for genomic targeting, leads to peri-implantation failure, overtly similar to *Setd1a* deficiency (Ref. 40).

This current manuscript uses siRNA knock-downs in MII oocytes to ablate mRNAs for MLL2, SETD1A and SETD1B, and/or to inhibit translation of these maternal transcripts. For SETD1A/B, the analysis is performed on a combined knock-down, because the authors detected up-regulation of *Setd1b* mRNA in 2-cell embryos with the *Setd1a* knock-down. The authors explore the effect of these knock-downs on developmental progression to blastocysts; on transcription by RNA-seq in 2-cell embryos, morulae and blastocysts; and H3K4me3 distribution by CUT&Tag. In parallel, they also assess the role of the H3K4 demethylase KDM5B by similar knock-down and over-expression. This represents a pretty comprehensive analysis of the molecular outcomes of the MLL2, SETD1A and SETD1B knock-downs, leading the authors to conclude, amongst other things, that MLL2 has the major role in deposition of canonical and non-canonical H3K4me3 at the earlier, totipotent, stages, and appears to be largely dispensable for gene expression control; whereas SETD1A/B catalyses canonical H3K4me3 deposition, and has a greater impact on gene transcription as preimplantation development progresses.

I believe the molecular analysis is performed to a high level. My chief concerns with the study are on the efficacy of the knock-downs, as we are provided with limited information on this key point.

Key questions:

1. What impact does the KD have on MLL2 protein level? Without knowing this information, the attribution of any effect observed is an inference. The only validation that the MLL2 knock-down was effective is what appears to be RT-qPCR in 2-cell embryos (Fig. S1D, but note that it is not specified whether the data shown are RT-qPCR or from RNA-seq). MLL2 is expressed and functional in the growing oocyte, so the extent to which the siRNA knock-down alters MLL2 protein levels is not known, given that there may be MLL2 provided from the oocyte which may persist for an unknown period during preimplantation development. This uncertainty is a real problem, and especially important because the results seem to suggest MLL2 is largely dispensable, with no effect on developmental progression, and RNA-seq analysis suggests minimal changes in gene expression (e.g., Fig. S6).

2. Likewise, in relation to the function of KDM5B over preimplantation stages, there is no demonstration of KDM5B protein level changes in the over-expression model; again, this is relevant because the effect of *Kdm5b* over-expression on transcription seems to be negligible according to RNA-seq analysis.

3. How effective and sustained are the knock-downs of SETD1A and SETD1B? The authors show Western blots for SETD1A and SETD1B proteins in late 2-cell embryos with the combined knock-down. Clearly, this is challenging technically, as each Western blot has required 200 manipulated embryos for a signal. The Western blots indicate some reduction in level of protein, but it is probably unfeasible to quantify this robustly, as there may not be replicates. Partial knock-downs - e.g., of the order of ~50% - may not be biologically significant (there is no evidence of haploinsufficiency for the two genes). Also, as many of the molecular endpoints are measured in morulae or blastocysts, do the authors have evidence that SETD1A/B protein or mRNA levels remain knocked-down at these stages? Some of this could be addressed in the RNA-seq datasets and by immunofluorescence analysis.

4. Generally, the RNA-level validation of the knock-downs in 2-cell embryos shown in Fig S1D-F, if these data are from RT-qPCR, should be validated in the corresponding RNA-seq datasets. RNA-seq should provide a much more robust quantification, as any fold-change of the transcript of interest can be normalised across the whole transcriptome, rather than against a single control gene.

Other comments and corrections:

Line 81-82: "We discerned that while MLL2 initiates H3K4me3 deposition pre- and during zygotic genome activation". MLL2 already deposits H3K4me3 in the growing oocyte, as in References 10 & 33. Please correct.

Lines 242-245: "Kdm5b overexpression significantly reduced non-canonical H3K4me3 at gene 242 promoters". By definition, non-canonical H3K4me3 is not at gene promoters. Please correct.

Lines 362-364: "These findings suggest that the loss of Setd1a/b" and "To determine whether the loss of Setd1a/b regulates gene expression through H3K4me3". Please rephrase. "Loss of" could be interpreted to mean complete absence, whereas the authors have been able to cause a reduction in abundance of SETD1A and SETD1B protein levels but not complete absence (Fig. S1G).

Line 401: "In the absence of Kdm5b". Again, this statement is too strong, it is not clear what the degree of reduction in KDM5B protein or Kdm5b mRNA has been achieved by the siRNA knock-down. The authors should instead say "In the Kdm5b knock-down" as well as providing some demonstration of the degree of knock-down achieved.

Lines 440-441: "In early mammalian embryos, non-canonical H3K4me3 associated with transcriptional silencing appears in embryos before ZGA". The authors should note that non-canonical H3K4me3 distribution is not universal amongst mammalian oocytes and preimplantation embryos; significantly, human oocytes/embryos do not have a non-canonical distribution of H3K4me3 (PMID: 34818044).

Lines 444-446: "we have revealed that MLL2 is responsible for the establishment and maintenance of nH3K4me3 before ZGA. The H3K4me3 mediated by MLL2 operates in a transcription-independent manner". The authors should reiterate here that MLL2 has already been identified as depositing H3K4me3 non-canonically in mouse oocytes, and doing so independent of transcription (PMID: 29323282).

Lines 452-455: "Early immunostaining studies indicated that the absence of nH3K4me3, resulting from Mll2 knockout or Kdm5b overexpression, disrupts genomic silencing in mature oocytes (11,33). However, later analyses using H3K4me3 ChIP-seq and RNA-seq in Mll2 knockout oocytes showed that the lack of nH3K4me3 doesn't directly impede transcription (10)". RNA-seq is not able to assess the process of transcription, only the mRNA population. Therefore, the RNA-seq analysis in Ref. 10 cannot be used as evidence to contradict the earlier work (Ref. 33) that used Brd-UTP incorporation as a direct measure of transcriptional activity, which showed that Mll2-KO oocytes do not undergo global repression of transcription. Please therefore correct the statement.

Lines 474-476: "simultaneous knockdown of both Setd1a and Setd1b significantly reduces the rate of blastocyst formation, a phenotype consistent with the results observed in Wdr82 knockdown and Cxxc1 knockout (24,40)". It is not clear that the information in the Cxxc1-knockout study (Ref. 40) is sufficient to be 'consistent' with reduced rate of blastocyst formation. Ref. 40 shows that Cxxc1^{-/-} embryos display early post-implantation failure. Blastocysts outgrowths were able to be obtained from Cxxc1^{-/-} embryos, but it is not clear that the numbers in the study are able to show whether there is significant deficit in blastocyst rate of the homozygotes, and the authors of that study do not claim this to be the case.

Lines 493-495: "Additionally, MLL2 is likely directed by ZGA-specific factors to CpG-sparse regions to establish Late2C-specific H3K4me3 peaks, a phenomenon that merits further exploration."

Did the authors explore the CpG density in H3K4me3 peaks? Note that in Ref. 10, it was found that the strongest determinant of MLL2 targeting in mouse oocytes was CG density. Non-canonical, MLL2-dependent H3K4me3 peaks may exist outside of the regions of greatest CG density (i.e., CpG island promoters), but CG density appears still to be an over-riding factor in genomic targeting of MLL2 at these stages.

Referee #3:

The manuscript by Jingjing Zhang et al addresses aspects of H3K4me3 epigenetics in the mouse from oogenesis to blastocyst with a focus on zygotic genome activation (ZGA) at the late 2C stage. This is interesting territory, which was initiated by the unexpected and counter-intuitive finding that H3K4me3 deposited by MLL2 during oogenesis relates to transcriptional silencing (and is required for safe transition through meiosis and subsequent oocyte quiescence) Andreu-Vieyra et al, 2010, ref 33, rather than the central role of H3K4me3 at active promoters. This breakthrough finding that H3K4me3 is associated with transcriptional silencing in the oocyte (here termed non-canonical H3K4me3) is not acknowledged in the manuscript. Also the previous recognition that the transition of H3K4me3 to association with active promoters (here termed canonical H3K4me3) is mediated by SETD1A (Bledau et al, ref 39) is also not acknowledged. However the authors focus their siRNA knock-down studies on these factors, additionally finding that Setd1b also needs to be co-knocked-down with Setd1a to reduce H3K4me3. The authors present some interesting findings that could be valuable contributions to this specialty interest.

However, before proceeding to comment on the text of this manuscript, two practical issues need to be raised.

1. The authors use siRNAs to achieve knock-downs either in MII oocytes or zygotes (for analyses in early 2C, late 2C, morula or blastocysts). It is widely known that RNAi off target effects are common, which has led to the establishment of required controls to demonstrate specificity. The controls usually employed either use two independent RNAi approaches (here that would be two completely different pools of 3 siRNAs directed against either Mll2, Setd1a/b or Kdm5b) or the use of wt rescue mRNAs that are resistant to the RNAi knock-down (or better both controls). Here the authors use a rescue for their Kdm5b knock-down but there are no controls for either Mll2 or Setd1a/b. Controls for RNAi knock-down specificity is a minimal requirement for publication in a quality journal.

2. The authors perform their knock-downs in either MII oocytes or zygotes but do not make it clear throughout the manuscript which results arise from MII oocytes and which from zygotes. This makes the work presented extremely difficult to evaluate, especially considering previously published work using conditional and developmental knock-outs of Mll2, Setd1a and Setd1b. Possibly some of the results presented here may confirm, or stand in contradiction to, the previously published work. Consequently, the statement in the abstract "Our research reveals that MLL2 is responsible for depositing H3K4me3 in totipotent embryos, with a transition to SETD1A/B in pluripotent embryos" could either be partially justifiable (it is already known that SETD1A is the major H3K4me3 in pluripotent embryos) or completely unjustified (because it is already known that MLL2 is the major H3K4me3 in late oocytes. Also SETD1B is required for late oogenesis and progress of the zygote). Notably none of these published findings (refs 33, 39, Brici et al, Development, 2017) are appropriately discussed because the authors do not compare their results with expectations arising from the existing state of knowledge.

The manuscript includes various misunderstandings.

a) line 62, in their demonstration that MLL2 is responsible for most H3K4me3 deposition on bivalent promoters in ESCs, Denissov et al (ref 34) also showed that MLL2 is found on most active (as well as bivalent) promoters. So the statement "whereas MLL1 and MLL2 target developmental gene promoters in a gene specific manner 15,16" is not correct. Ref 16 does not claim to show that MLL1 targets developmental gene promoters in a gene specific manner. MLL1, as does MLL2, appears to bind to most active promoters e.g. Guenther et al, PNAS 102, 8603 2005).

b) line 72-74. "Previous research indicates that the deficiency of any histone lysine methyltransferase results in embryonic fatality between E7.5 and E11.5 12" is not correct. Presumably this statement refers to H3K4 lysine methyltransferases and not all the other lysine methyltransferase as well. Even then it is not correct. Setd1a knock-outs stop at E5.5 and do not die at E7.5 (ref 39) and MLL3 knock-outs proceed to birth (Jang et al, 2013).

c) Figure 1B, presumably the 55,335 peaks refers to all H3K4me3 peaks and not 'All genes'. The mouse genome has 22,000 genes and about half are expressed in early development, so the figure legend (line 140) 'Each row represents a promoter region (TSS +/- 2 kb) cannot be correct.

c) line 185/6 and Figure 2D. 'Significantly, transcriptional inhibition selectively reduced H3K4me3 in morulae on shared peaks ...' - is very hard to see on Fig. 2D.

d) line 199, Figure 2 legend. 'Poll52' - there is no reference 52.

e) line 451 - 456. 'Despite initial findings linking nH3K4me3 in mouse oocytes and pre-ZGA embryos with transcriptional silence, its role in transcriptional suppression remains contentious.....However, later analyses using H3K4me3 ChIP-seq and RNA-seq knockout oocytes showed that the lack of nH3K4me3 doesn't directly impede transcription¹⁰'. These statements are not correct. First, the linkage between nH3K4me3 and transcriptional silencing is not contentious (however the molecular explanation remains open). Second, the 'later analyses showed that the lack of showed that the lack of nH3K4me3 doesn't directly impede transcription¹⁰' doesn't directly impede transcription¹⁰'. But this has nothing to do with global transcriptional silencing which correlates with nH3K4me3. This confusing illogicality is used by the authors to establish unjustified priority for their findings. It is followed in the Discussion by a series of unreasonable contentions (lines 458 - 462).

f) line 462 - 464 'The absence of H3K27me3 and DNA methylation in early embryos indicates that MLL2's role in early embryogenesis is distinct from it's function in ESCs'. This is statement is not only illogical (how does 'the absence of H3K27me3 and DNA methylation in early embryos' indicate anything about MLL2?) but also fails to acknowledge that it has already been published that 'MLL2's role in early embryogenesis is distinct from it's function in ESCs' (and not cited).

Additional comments -

1. The authors generally equate loss of MLL2 or SETD1A/B to reduced H3K4me3 without considering that these very large proteins are embedded in very large complexes of at least 8 other proteins. They are not simply H3K4 methyltransferase and all these proteins have many other functions that these knock-downs also perturb.

2. The authors perpetuate the confusion accompanying the term 'COMPASS', which means Complex associated with Set1. Mammals have six H3K4 methyltransferase complexes only two of which are associated with SET1. The other four mammalian complexes have substantial differences. The core problem here is that the term 'COMPASS' is imprecise. It is not correct to call all H3K4 methyltransferase complexes 'COMPASS' in the same way it would not be correct to call MLL1, MLL2, MLL3 or MLL4 'SET1'. Line 59 is a relevant example of this confusion where the six separate mammalian complexes are bundled together under the term 'COMPASS complex'. Whereas insiders in this field are used to this problem, there is no justifiable reason to keep perpetuating this confusion. I recommend 'SET1/MLL complexes', which has utility and is more precise.

Referee #1:

The authors focused on the biological significance of non-canonical (nc) and canonical (c) H3K4me3 distribution as well as its transition. Mll2 dictated cH3K4me3 distribution in a transcription-independent manner. In contrast, Setd1a/b maintained cH3K4me3 distribution in a transcription-dependent manner. Interestingly, Pol II inhibition by Triptolide for 2h was sufficient to reduce cH3K4me3 enrichment in morula embryos. Additional analyses revealed that the reduction of nc/cH3K4me3 enrichment by Mll2 KD did not affect the transcriptome of 2-cell embryos, which was also the case in Kdm5b-overexpressed embryos. Thus, global reduction of H3K4me3 did not affect ZGA. Next, the authors instead performed Kdm5b KD and observed the enhancement of ncH3K4me3 and cH3K4me3 enrichment in late 2-cell embryos. However, there was no significant correlation between changes in H3K4me3 and the alteration of gene expression. Thus, reduced as well as enhanced H3K4me3 enrichment in 2-cell embryos did not affect ZGA. In contrast, Setd1a/b KD impaired development to the blastocyst stage and affected lineage specification. Notably, decrease in Setd1a/b-dependent H3K4me3 was linked to the down-regulation of gene expression including Pou5f1, Nanog, and Cdx2. Kdm5b KD also impaired development to the blastocyst stage. However, in this case, Nanog-positive cells increased. Inhibitor assays showed that Kdm5b activity is required after the late 2-cell stage rather than the earlier period.

This paper enthusiastically addressed the biological significance of H3K4me3 in early mouse embryos. Overall, the authors' data provided are convincing, and the provided results support the authors' conclusion. Thus I support the publication of this paper in EMBO J, however, I suggest the authors to address the following points before the publication.

Response: We sincerely appreciate the positive evaluation provided by the referee regarding our work.

1. Fig. 1B,C,2D - Show changes in H3K4me3 enrichment at genome-wide bins as shown in Fig S2B, as it is unclear whether Mll2 KD or Setd1a/b KD causes only the global reduction of H3K4me3. I assume that H3K4me3 at some genomic regions are resistant to these KD. To address this point, please show which regions are preferentially affected or unaffected.

Response: Thank you for your valuable suggestion. Following your recommendation, we initially illustrated the changes in H3K4me3 enrichment across 5-kb genome-wide bins after *Mll2* KD or *Setd1a/b* KD, as shown in Fig S2B. Our findings demonstrate a genome-wide loss of H3K4me3 levels in Late2C embryos following *Mll2* KD and in morulae after *Setd1a/b* KD (Figure R1A). We then identified bins with differential H3K4me3 enrichment applying a two-fold change threshold. Specifically, *Mll2* KD resulted in a reduction in H3K4me3 enrichment in 70% of the bins with no increase observed in Late2C embryos, while *Setd1a/b* KD led to a decrease in 3% and an increase in 1% of the bins (Figure R1B). Conversely, in morulae, *Mll2* KD reduced H3K4me3 enrichment in 2% of the bins and increased it in 1%, whereas *Setd1a/b* KD led to a decrease in 42% of the bins, with no increases noted (Figure R1B). We have added these results in Appendix Figures S2E and S2F. We hope these additions effectively address your concerns.

Figure R1. Genome-wide H3K4me3 enrichment dynamics after *Mll2* or *Setd1a/b* KD. **(A)** Scatter plots depict alterations in genome-wide H3K4me3 enrichment levels within 5-kb bins following *Mll2* KD or *Setd1a/b* KD at the Late2C and morula stages. **(B)** Bar charts display the distribution of 5-kb H3K4me3 bins across different categories of fold change following *Mll2* or *Setd1a/b* KD at the Late2C and morula stages.

2. Fig. 2C,D -It is intriguing that H3K4me3 disappears within 2h treatment of Trp in morulae. However, further analysis was not performed in the current manuscript. I strongly suggest the authors to address the underlying mechanism of this phenomenon with additional experiments. Although Pol II-mediated recruitment of *Setd1a/b* may be operative (as described in Discussion), this discussion only would not be enough to explain this observation. For example, the authors can address the involvement of KDMs in the reduction of H3K4me3 caused by Trp.

Response: Thank you very much for your constructive suggestion. Following your insightful advice, we further explored the potential role of KDM5s in the Trp-induced reduction of H3K4me3. Accordingly, we conducted additional H3K4me3 immunostaining experiments on morulae, both with Trp treatment alone and in combination with the KDM5s inhibitor CPI-455. Our results indicate that when combined with CPI-455, the decrease in H3K4me3 levels induced by Trp alone is partially mitigated (Figure R2). This suggests that KDM5s play a significant role in the reduction of H3K4me3 following Trp exposure in morulae. We have added these results in Appendix Figures S5F, detailed on lines 195-200 of the revised manuscript. Additionally, we have updated the discussion in lines 440-450 of the revised manuscript.

Figure R2. Immunostaining shows H3K4me3 (green) and DNA (gray) in morulae treated with Triptolide alone or in combination with CPI-455 for 0, 2, 4, and 6 hours on the top, alongside a quantitative analysis of H3K4me3 relative intensities on the bottom panel. Scale bar, 20 μ m. Error bars represent SD. ** $P < 0.01$; *** $P < 0.001$; two-sided unpaired Student's *t*-test.

3. Fig. 3 - The authors' data showing H3K4me3 is dispensable for major ZGA is convincing. However, it is unclear whether *Kdm5b* OE promotes the non-canonical-to-canonical transition precociously or just globally reduce the enrichment of H3K4me3. This investigation would be important to interpret the data.

Response: Thanks for your good suggestion. In the revised manuscript, we assessed the impact of *Kdm5b* overexpression on canonical H3K4me3 peaks. The results indicated that *Kdm5b* overexpression does not induce a premature formation of canonical H3K4me3 in Early2C embryos (Figure R3). This suggests that while *Kdm5b* overexpression accelerates the clearance of non-canonical H3K4me3 in Early2C embryos, it does not promote the transition from non-canonical to canonical forms. We have added these results in Appendix Figures S8D, detailed on lines 235-236 of the revised manuscript.

Figure R3. Heatmap showing H3K4me3 enrichment levels within cH3K4me3 peaks ($n = 65,898$) in Early2C embryos (both *Kdm5b* WT and *Kdm5b* MUT) and Late2C embryos.

4. Fig. 4F - Please show the commonality of DEGs between KD and inhibitor-treated samples by venn diagrams.

Response: We have now incorporated two Venn diagram to delineate the relationships and overlaps between DEGs in the *Kdm5b* KD and the CPI-455 treated groups. Specifically, there is an overlap of 66 down-regulated genes and 23 up-regulated genes between the *Kdm5b* KD and CPI-455 treated groups. We hope our additional results will meet your requirements.

Figure R4. Venn diagram illustrates the intersection of DEGs between control group and *Kdm5b* KD group, as well as between DMSO-treated and CPI-455-treated groups in Late2C embryos.

5. Fig. S10 - Effect of *Kdm5b* KD on H3K4me3 and development is highly convincing as the authors provide the results of rescue experiments. I strongly suggest the authors to perform H3K4me3 CUT&TAG and RNA-seq using *Kdm5b* KD morula samples. This will reveal how the absence of *Kdm5b* alters H3K4me3 distribution and whether enhanced H3K4me3 enrichment directly affects gene expression. I believe that this is a bit tough experiment for a revision but is required to meet the criteria of EMBO J.

Response: Thank you for your constructive suggestion. Following your suggestions, we conducted H3K4me3 CUT&Tag and RNA-seq analyses in control and *Kdm5b* KD morula samples. Our findings demonstrate that *Kdm5b* KD significantly elevated H3K4me3 enrichment at gene promoters and broadened H3K4me3 peaks (Figure R5A, B). Furthermore, *Kdm5b* KD

resulted in the up-regulation of 264 genes and down-regulation of 241 genes (Figure R5C-E). Gene Ontology analysis revealed that up-regulated genes predominantly concentrated in the regulation of epithelial to mesenchymal transition and embryonic foregut morphogenesis; the down-regulated genes were largely associated with placental development, aligning with the observed phenotype of diminished TE cell counts in blastocysts (Figure R5F). Additionally, genes such as *Pou5f1*, *Fgf4*, and *Gata4*, which were up-regulated following *Kdm5b* KD, showed increased H3K4me3 enrichment and broader peaks at their promoters (Figure R5G, H). These results underscore the critical role of KDM5B in modulating the expression of pivotal genes that govern early cell fate determination by finely tuning H3K4me3 dynamics. We have added these results in Figure 6 and Appendix Figures S13, detailed on lines 345-356 of the revised manuscript.

Figure R5. Knockdown of *Kdm5b* leads to aberrant distribution of H3K4me3 and dysregulation

of gene expression. (A) Scatter plots illustrate correlations between biological replicates of H3K4me3 CUT&Tag data in control and *Kdm5b* KD morulae. H3K4me3 enrichment was calculated as RPKM using 5-kb bins. Pearson correlation coefficients are shown on the top-left panel. (B) Box plot showing the H3K4me3 enrichment and peak length at the promoter regions (TSS \pm 2 kb) in control and *Kdm5b* KD morulae. *** $P < 0.001$; two-sided Wilcoxon-Mann-Whitney test. (C) Principal component analysis of gene expression in control and *Kdm5b* KD morulae. (D) Expression levels of *Kdm5b* at the morula stage following *Kdm5b* KD are assessed via RNA-seq. Adjusted P -values are computed using DESeq2. Error bars denote SD from three biological replicates. *** $P < 0.001$. LFC, Log₂ (fold change). (E) Scatter plots comparing the gene expression levels between control and *Kdm5b* KD morulae, highlighting genes up-regulated (FC > 2 and P adjusted < 0.05, red) and down-regulated in *Kdm5b* KD embryos (FC > 2 and P adjusted < 0.05, blue). (F) Bar charts showing the enriched GO terms for genes up-regulated and down-regulated in *Kdm5b* KD morulae. (G) Box plots showing the H3K4me3 enrichment levels at the promoter regions (TSS \pm 2 kb) of down-regulated, up-regulated, and all genes when comparing control with *Kdm5b* KD morulae. n.s., not significant; ** $P < 0.01$; *** $P < 0.001$; two-sided Wilcoxon-Mann-Whitney test. (H) Genome browser views showing the H3K4me3 enrichments for representative up-regulated genes in control and *Kdm5b* KD morulae.

6. Fig. 6E - Please include the illustration summarizing the results from *Kdm5b* OE and KD experiments.

Response: Thank you for your suggestion. We have included a summary of the results from the *Kdm5b* KD experiments in Figure 7 of the revised manuscript. As our *Kdm5b* OE experiments were limited to the Early2C stage, and we did not investigate the effects of *Kdm5b* OE on H3K4me3 at the Late2C and morula stages, these results were not incorporated into the schematic. We hope this addresses your query appropriately.

Figure R6. Models of the dynamic establishment and functions of H3K4me3 in mouse early embryos.

Referee #2:

This manuscript comprises a series of knock-downs and over-expression studies to examine the contributions of the H3K4 methyltransferases MLL2, and SETD1A and SETD1B in mouse preimplantation embryos, and how they contribute to the changes in deposition of H3K4me3, including the transition for 'non-canonical' (i.e., with pervasive broad, intergenic domains) to 'canonical' (at promoters and CpG islands), and the consequential impact on fidelity of zygotic genome activation. Prior work has shown that MLL2 is an essential factor for oocyte development and competence (Ref. 33), and is responsible for the deposition of 'non-canonical' H3Kme3 independent of transcription in oocytes (Ref. 10). Separate zygotic genetic knock-outs of *Setd1a* and *Setd1b* have shown that both knock-outs are compatible with preimplantation development to blastocysts, but *Setd1a* is essential for further development of ICM, whereas *Setd1b* becomes essential only later in development (Ref. 39). Notably, combined oocyte and zygotic knock-out of *Setd1a* was reported not to lead to a more severe development phenotype, suggesting that maternally provided SETD1A is dispensable in the preimplantation embryo. In comparison, the original genetic knock-out of *Cxxc1/Cfp1*, which is required by both SETD1A and SETD1B for genomic targeting, leads to peri-implantation failure, overtly similar to *Setd1a* deficiency (Ref. 40).

This current manuscript uses siRNA knock-downs in MII oocytes to ablate mRNAs for MLL2, SETD1A and SETD1B, and/or to inhibit translation of these maternal transcripts. For SETD1A/B, the analysis is performed on a combined knock-down, because the authors detected up-regulation of *Setd1b* mRNA in 2-cell embryos with the *Setd1a* knock-down. The authors explore the effect of these knock-downs on developmental progression to blastocysts; on transcription by RNA-seq in 2-cell embryos, morulae and blastocysts; and H3K4me3 distribution by CUT&Tag. In parallel, they also assess the role of the H3K4 demethylase KDM5B by similar knock-down and over-expression. This represents a pretty comprehensive analysis of the molecular outcomes of the MLL2, SETD1A and SETD1B knock-downs, leading the authors to conclude, amongst other things, that MLL2 has the major role in deposition of canonical and non-canonical H3K4me3 at the earlier, totipotent, stages, and appears to be largely dispensable for gene expression control; whereas SETD1A/B catalyses canonical H3K4me3 deposition, and has a greater impact on gene transcription as preimplantation development progresses.

I believe the molecular analysis is performed to a high level. My chief concerns with the study are on the efficacy of the knock-downs, as we are provided with limited information on this key point.

Response: We appreciate the referee's insightful summary and support for our work. We have supplemented the revised manuscript with detailed information on the knockdown efficiency to enhance the comprehensiveness and reliability of the study.

Key questions:

1. What impact does the KD have on MLL2 protein level? Without knowing this information, the attribution of any effect observed is an inference. The only validation that the MLL2 knock-down

was effective is what appears to be RT-qPCR in 2-cell embryos (Fig. S1D, but note that it is not specified whether the data shown are RT-qPCR or from RNA-seq). MLL2 is expressed and functional in the growing oocyte, so the extent to which the siRNA knock-down alters MLL2 protein levels is not known, given that there may be MLL2 provided from the oocyte which may persist for an unknown period during preimplantation development. This uncertainty is a real problem, and especially important because the results seem to suggest MLL2 is largely dispensable, with no effect on developmental progression, and RNA-seq analysis suggests minimal changes in gene expression (e.g., Fig. S6).

Response: Thank you for your thorough referee and helpful suggestions. We apologize for the oversight in not specifying the data source previously. In response, we have updated the figures and figure legends in the revised manuscript to clearly indicate that the data in Appendix Figures S1C-E is derived from RT-qPCR analyses. Moreover, the revised manuscript now includes RNA-seq data and fluorescence staining results, demonstrating that *Mll2* KD results in a substantial decrease (exceeding 75%) in both mRNA and protein levels of MLL2 in early embryos, as shown in Figure R7. These findings are detailed in Appendix Figure S6C for the RNA-seq data and Appendix Figure S1F for the immunostaining results.

Figure R7. Evaluation of *Mll2* knockdown efficiency. **(A)** Expression levels of *Mll1*, *Mll2*, *Setd1a*, and *Setd1b* at the Early2C and Late2C stages following *Mll2* KD are assessed via RNA-seq. Adjusted *P*-values are computed using DESeq2. Error bars denote SD from two biological replicates. n.s., no significance; ***P* < 0.01; ****P* < 0.001. LFC, Log₂ (fold change). **(B)** Immunostaining for MLL2 (red) and DNA (gray) in control and *Mll2* KD embryos at the Late2C and blastocyst stage is shown on the left. The right panel quantifies the relative intensities of MLL2. Scale bar, 20 μm. Error bars represent SD. ****P* < 0.001; two-sided Student's *t*-test.

2. Likewise, in relation to the function of KDM5B over preimplantation stages, there is no

demonstration of KDM5B protein level changes in the over-expression model; again, this is relevant because the effect of *Kdm5b* over-expression on transcription seems to be negligible according to RNA-seq analysis.

Response: We greatly appreciate your insightful comments and have addressed your concerns in the following manner: We evaluated the protein levels of KDM5B in Early2C embryos injected with *Kdm5b* mRNA through immunostaining. The results demonstrated that the levels of both wild-type (WT) and mutant (MUT) KDM5B proteins increased approximately 15-fold compared to the control group (Figure R8). To further substantiate the rigor of our findings, we have included additional immunostaining results for KDM5B in Appendix Figure S8A.

Figure R8. Immunostaining for KDM5B (red) and DNA (gray) in control, *Kdm5b* WT, and *Kdm5b* MUT embryos at the Early2C stage is shown on the left. The right panel quantifies the relative intensities of KDM5B. Scale bar, 20 μ m. Error bars represent SD. *** $P < 0.001$ two-sided Student's *t*-test.

3. How effective and sustained are the knock-downs of SETD1A and SETD1B? The authors show Western blots for SETD1A and SETD1B proteins in late 2-cell embryos with the combined knock-down. Clearly, this is challenging technically, as each Western blot has required 200 manipulated embryos for a signal. The Western blots indicate some reduction in level of protein, but it is probably unfeasible to quantify this robustly, as there may not be replicates. Partial knock-downs - e.g., of the order of ~50% - may not be biologically significant (there is no evidence of haploinsufficiency for the two genes). Also, as many of the molecular endpoints are measured in morulae or blastocysts, do the authors have evidence that SETD1A/B protein or mRNA levels remain knocked-down at these stages? Some of this could be addressed in the RNA-seq datasets and by immunofluorescence analysis.

Response: Thank you once again for your insightful suggestions. Accordingly, we have revised the manuscript to include an assessment of the mRNA and protein levels of SETD1A and SETD1B, utilizing RNA-seq data and immunostaining analysis, respectively. The findings indicate that *Setd1a* and *Setd1b* achieve a knockdown efficiency exceeding 70% at both the Late2C and blastocyst stages (Figure R9). This demonstrates that *Setd1a* and *Setd1b* can be effectively and sustainably knocked down in early embryos. These findings are detailed in Appendix Figure S10C for the RNA-seq data and Appendix Figure S1G,H for the immunostaining results.

Figure R9. Evaluation of knockdown efficiency for *Setd1a* and *Setd1b*. **(A)** Expression levels of *Mll1*, *Mll2*, *Setd1a*, and *Setd1b* at the morula and blastocyst stages following *Setd1a/b* KD are assessed via RNA-seq. Adjusted *P*-values are computed using DESeq2. Error bars denote SD from two biological replicates. n.s., no significance; **P* < 0.05; ***P* < 0.01. LFC, Log₂ (fold change). **(B)** Immunostaining for SETD1A (red) and DNA (gray) in control and *Setd1a/b* KD embryos at the Late2C and blastocyst stage is shown on the left. The right panel quantifies the relative intensities of SETD1A. Scale bar, 20 μm. Error bars represent SD. ****P* < 0.001; two-sided Student's *t*-test. **(C)** Immunostaining for SETD1B (red) and DNA (gray) in control and *Setd1a/b* KD embryos at the Late2C and blastocyst stage is shown on the left. The right panel quantifies the relative intensities of SETD1B. Scale bar, 20 μm. Error bars represent SD. ****P* < 0.001; two-sided Student's *t*-test.

4. Generally, the RNA-level validation of the knock-downs in 2-cell embryos shown in Fig S1D-F, if these data are from RT-qPCR, should be validated in the corresponding RNA-seq datasets. RNA-seq should provide a much more robust quantification, as any fold-change of the transcript of interest can be normalised across the whole transcriptome, rather than against a single control gene.

Response: Following your recommendations, we have validated the RNA levels depicted in

Figures S1D-F using the existing RNA-seq dataset. Figure R7A displays the expression levels of *Mll1*, *Mll2*, *Setd1a*, and *Setd1b* subsequent to *Mll2* KD at the Early 2C and Late 2C stages. Similarly, Figure R9A illustrates the expression levels of these genes post *Setd1a/b* KD at the morula and blastocyst stages. The RNA-seq results for *Mll2* KD are elaborated in Appendix Figure S6C, and those for *Setd1a/b* KD are detailed in Appendix Figure S10C. Furthermore, we present RNA-seq data for *Kdm5b* expression in control and *Kdm5b* KD embryos at the Late2C and morula stages, as detailed in Appendix Figures S9I and S13G.

Other comments and corrections:

Line 81-82: "We discerned that while MLL2 initiates H3K4me3 deposition pre- and during zygotic genome activation". MLL2 already deposits H3K4me3 in the growing oocyte, as in References 10 & 33. Please correct.

Response: Thank you for your valuable feedback. We acknowledge that MLL2 does not initiate the deposition of H3K4me3 in embryos but rather earlier in growing oocytes. Consequently, we have revised the sentence to: "We discerned that while MLL2 is responsible for the H3K4me3 deposition pre- and during zygotic genome activation (ZGA), SETD1A and SETD1B subsequently assume this role in a redundant manner following the initiation of ZGA" as stated in the revised manuscript (Lines 89-91).

Lines 242-245: "Kdm5b overexpression significantly reduced non-canonical H3K4me3 at gene 242 promoters". By definition, non-canonical H3K4me3 is not at gene promoters. Please correct.

Response: Thank you for your attention to detail in reviewing our manuscript. The non-canonical H3K4me3 was initially identified by Professor Wei Xie's laboratory at Tsinghua University. They described it as follows: "Notably, even at promoters, the pattern of ncH3K4me3 in MII oocytes is different from canonical H3K4me3. Promoter ncH3K4me3 is preferentially enriched in the low-CpG regions instead of high-CpG regions" (Zhang, Zheng et al., 2016). In the review from this laboratory, it is also noted that non-canonical ncH3K4me3 can be localized to promoters: "For example, at later stages of oocyte growth (growing oocyte II), H3K4me3 tends to appear as a broad domain at both promoters and distal sites, including many intergenic regions, forming a noncanonical pattern of H3K4me3 (ncH3K4me3)" (Xu & Xie, 2018). We hope this clarification addresses your concerns.

References:

Xu Q, Xie W (2018) Epigenome in Early Mammalian Development: Inheritance, Reprogramming and Establishment. *Trends Cell Biol* 28: 237-253

Zhang B, Zheng H, Huang B, Li W, Xiang Y, Peng X, Ming J, Wu X, Zhang Y, Xu Q, Liu W, Kou X, Zhao Y, He W, Li C, Chen B, Li Y, Wang Q, Ma J, Yin Q et al. (2016) Allelic reprogramming of the histone modification H3K4me3 in early mammalian development. *Nature* 537: 553-557

Lines 362-364: "These findings suggest that the loss of *Setd1a/b*" and "To determine whether the loss of *Setd1a/b* regulates gene expression through H3K4me3". Please rephrase. "Loss of" could be interpreted to mean complete absence, whereas the authors have been able to cause a reduction in abundance of SETD1A and SETD1B protein levels but not complete absence (Fig. S1G).

Response: Thank you for your meticulous attention to detail. We have amended the sentence to: "These findings suggest that the reduction in *Setd1a/b* leads to the down-regulation of genes essential for the cell fate commitment" as stated in the revised manuscript (Lines 303-305). Furthermore, we have thoroughly reviewed the revised manuscript and adjusted instances of the terms such as "loss", "lack", and "absence" to "reduction" or "knockdown" to ensure clarity and precision in our terminology.

Line 401: "In the absence of *Kdm5b*". Again, this statement is too strong, it is not clear what the degree of reduction in KDM5B protein or *Kdm5b* mRNA has been achieved by the siRNA knock-down. The authors should instead say "In the *Kdm5b* knock-down" as well as providing some demonstration of the degree of knock-down achieved.

Response: Thank you for your valuable feedback. We have amended line 401 to "In the *Kdm5b* knockdown" to more accurately reflect the experimental conditions. In addition to the RT-qPCR data, we have incorporated RNA-seq and immunostaining in the revised manuscript (Figure R10). These additions provide a more comprehensive demonstration of the siRNA knockdown's effect on KDM5B protein and *Kdm5b* mRNA levels, further clarifying knockdown efficiency. These findings are detailed in Appendix Figures S9I and S13G for the RNA-seq data, and in Appendix Figure S9B for the immunostaining results.

Figure R10. Evaluation of knockdown efficiency for *Kdm5b*. (A) Bar chart showing the relative expression levels of *Kdm5a* and *Kdm5b* after knocking down of *Kdm5b* or *Kdm5a/b* at the Late2C

stage, as detected by RT-qPCR. Error bars represent SD from three biological replicates. n.s., no significance; * $P < 0.05$; two-sided Student's t -test. **(B)** Expression levels of *Kdm5b* at the Late2C and morula stages following *Kdm5b* KD are assessed via RNA-seq. Adjusted P -values are computed using DESeq2. Error bars denote SD from three biological replicates. *** $P < 0.001$. LFC, Log_2 (fold change). **(C)** Immunostaining for KDM5B (red) and DNA (gray) in control and *Kdm5b* KD embryos at the Late2C and blastocyst stage is shown on the left. The right panel quantifies the relative intensities of KDM5B. Scale bar, 20 μm . Error bars represent SD. *** $P < 0.001$; two-sided Student's t -test.

Lines 440-441: "In early mammalian embryos, non-canonical H3K4me3 associated with transcriptional silencing appears in embryos before ZGA". The authors should note that non-canonical H3K4me3 distribution is not universal amongst mammalian oocytes and preimplantation embryos; significantly, human oocytes/embryos do not have a non-canonical distribution of H3K4me3 (PMID: 34818044).

Response: We have carefully reviewed the reference you provided and found that the study supports the presence of non-canonical H3K4me3 in oocytes and embryos across non-human species, aligning with our observations in early pig embryos (Bu, Zhu et al., 2022). While non-canonical H3K4me3 is absent in human oocytes, it does appear transiently in human embryos prior to zygotic genome activation (ZGA), as depicted in the schematic diagram from the cited reference. We hope this addresses your concerns.

Figure R11. Schematic diagram sourced from the reference (Lu, Zhang et al., 2021).

Reference:

Bu G, Zhu W, Liu X, Zhang J, Yu L, Zhou K, Wang S, Li Z, Fan Z, Wang T, Hu T, Hu R, Liu Z, Wang T, Wu L, Zhang X, Zhao S, Miao YL (2022) Coordination of zygotic genome activation entry and exit by H3K4me3 and H3K27me3 in porcine early embryos. *Genome Res* 32: 1487-1501

Lu X, Zhang Y, Wang L, Wang L, Wang H, Xu Q, Xiang Y, Chen C, Kong F, Xia W, Lin Z, Ma S, Liu L, Wang X, Ni H, Li W, Guo Y, Xie W (2021) Evolutionary epigenomic analyses in mammalian early embryos reveal species-specific innovations and conserved principles of imprinting. *Sci Adv* 7: eabi6178

Lines 444-446: "we have revealed that MLL2 is responsible for the establishment and maintenance of ncH3K4me3 before ZGA. The H3K4me3 mediated by MLL2 operates in a transcription-independent manner". The authors should reiterate here that MLL2 has already been identified as depositing H3K4me3 non-canonically in mouse oocytes, and doing so independent of transcription (PMID: 29323282).

Response: Thank you for your observation. We have amended the sentence to: "we discovered that MLL2 deposits ncH3K4me3 independently of transcription before ZGA, consistent with previous findings in mouse oocytes(Hanna et al., 2018)" as stated in the revised manuscript (Lines 365-367).

Lines 452-455: "Early immunostaining studies indicated that the absence of ncH3K4me3, resulting from Mll2 knockout or Kdm5b overexpression, disrupts genomic silencing in mature oocytes (11,33). However, later analyses using H3K4me3 ChIP-seq and RNA-seq in Mll2 knockout oocytes showed that the lack of ncH3K4me3 doesn't directly impede transcription (10)". RNA-seq is not able to assess the process of transcription, only the mRNA population. Therefore, the RNA-seq analysis in Ref. 10 cannot be used as evidence to contradict the earlier work (Ref. 33) that used Brd-UTP incorporation as a direct measure of transcriptional activity, which showed that Mll2-KO oocytes do not undergo global repression of transcription. Please therefore correct the statement.

Response: Thank you for your insightful correction. We acknowledge that the RNA-seq analysis results presented in Reference 10 cannot be directly used to counter the earlier findings from Reference 33, which were derived using the Brd-UTP incorporation method. We have removed this erroneous statement in the revised manuscript.

Lines 474-476: "simultaneous knockdown of both Setd1a and Setd1b significantly reduces the rate of blastocyst formation, a phenotype consistent with the results observed in Wdr82 knockdown and Cxxc1 knockout (24,40)" It is not clear that the information in the Cxxc1-knockout study (Ref. 40) is sufficient to be 'consistent' with reduced rate of blastocyst formation. Ref. 40 shows that Cxxc1^{-/-} embryos display early post-implantation failure. Blastocysts outgrowths were able to be obtained from Cxxc1^{-/-} embryos, but it is not clear that the numbers in the study are able to show whether there is significant deficit in blastocyst rate of the homozygotes, and the authors of that study do not claim this to be the case.

Response: Thank you for your comprehensive review and insightful correction. We recognize that Reference 40 does not furnish data supporting the claim that *Cxxc1* knockout results in a reduced rate of blastocyst formation. Consequently, we have excised the mention of *Cxxc1* knockout along with the associated reference from the revised manuscript.

Lines 493-495: "Additionally, MLL2 is likely directed by ZGA-specific factors to CpG-sparse regions to establish Late2C-specific H3K4me3 peaks, a phenomenon that merits further exploration." Did the authors explore the CpG density in H3K4me3 peaks? Note that in Ref. 10, it was found that the strongest determinant of MLL2 targeting in mouse oocytes was CG density. Non-canonical, MLL2-dependent H3K4me3 peaks may exist outside of the regions of greatest CG density (i.e., CpG island promoters), but CG density appears still to be an over-riding factor in genomic targeting of MLL2 at these stages.

Response: Thank you for reviewer valuable comment highlighting the role of CpG density in MLL2 targeting. In our current study, we have specifically addressed this point by investigating the CpG density of Late2C-specific, Morula-specific, and shared H3K4me3 peaks. As detailed in our analyses and illustrated in Figure 2A, our findings demonstrate a preference for lower CpG density in Late2C-specific and Morula-specific peaks compared to shared peaks. Furthermore, we observed that H3K4me3 peaks in high CpG density areas at the Late2C stage tend to accumulate further at the morula stage. This aspect of our study suggests that while CpG density is a primary determinant, other regulatory factor may also contribute to the establishment of MLL2-dependent H3K4me3 peaks in regions with varying CpG densities. We hope this addresses your concerns.

Figure R12. Displays heatmaps of H3K4me3, Pol II, and CpG density for Late2C-specific, shared, and morula-specific H3K4me3 peaks, along with Gene Ontology (GO) terms. Rows are sorted by descending H3K4me3 enrichment (RPKM value), with two replicates per sample.

Referee #3:

The manuscript by Jingjing Zhang et al addresses aspects of H3K4me3 epigenetics in the mouse from oogenesis to blastocyst with a focus on zygotic genome activation (ZGA) at the late 2C stage. This is interesting territory, which was initiated by the unexpected and counter-intuitive finding that H3K4me3 deposited by MLL2 during oogenesis relates to transcriptional silencing (and is required for safe transition through meiosis and subsequent oocyte quiescence) Andreu-Vieyra et al, 2010, ref 33, rather than the central role of H3K4me3 at active promoters. This breakthrough finding that H3K4me3 is associated with transcriptional silencing in the oocyte (here termed non-canonical H3K4me3) is not acknowledged in the manuscript. Also the previous recognition that the transition of H3K4me3 to association with active promoters (here termed canonical H3K4me3) is mediated by SETD1A (Bledau et al, ref 39) is also not acknowledged. However the authors focus their siRNA knock-down studies on these factors, additionally finding that *Setd1b* also needs to be co-knocked-down with *Setd1a* to reduce H3K4me3.

The authors present some interesting findings that could be valuable contributions to this specialty interest.

Response: We thank the referee for their detailed assessment of our study and their constructive feedback, which will guide the enhancement of our manuscript.

However, before proceeding to comment on the text of this manuscript, two practical issues need to be raised.

1. The authors use siRNAs to achieve knock-downs either in MII oocytes or zygotes (for analyses in early 2C, late 2C, morula or blastocysts). It is widely known that RNAi off target effects are common, which has led to the establishment of required controls to demonstrate specificity. The controls usually employed either use two independent RNAi approaches (here that would be two completely different pools of 3 siRNAs directed against either *Mll2*, *Setd1a/b* or *Kdm5b*) or the use of wt rescue mRNAs that are resistant to the RNAi knock-down (or better both controls). Here the authors use a rescue for their *Kdm5b* knock-down but there are no controls for either *Mll2* or *Setd1a/b*. Controls for RNAi knock-down specificity is a minimal requirement for publication in a quality journal.

Response: We appreciate the referee's valuable suggestions and fully recognize the importance of verifying siRNA specificity. To address this issue, we initially designed new siRNAs targeting the UTR regions of *Setd1a* and *Setd1b*, followed by rescue experiments involving overexpression of wild-type (WT) CDS-only mRNA. However, none of the 12 siRNAs targeting the UTR region effectively reduced *Setd1a* expression.

Given the high degree of conservation of SETD1A/B in mammals, evidenced by over 88% protein sequence similarity between human and mouse (Figure R13), and the capability of human SETD1A protein to compensate for mouse *Setd1a* deficiency (Hoshii, Cifani et al., 2018), we continued employing the original mouse siRNAs, which do not target human *SETD1A* and *SETD1B* (Figure R14A), to knock down mouse *Setd1a* and *Setd1b*, and subsequently overexpressed human *SETD1A* and *SETD1B* mRNA (Figure R14B-D). The results indicated that human WT *SETD1A/B* expression (hWT) successfully rescued the development of *Setd1a/b*

knockdown embryos (Figure R14E,F). Additionally, we engineered catalytically inactive human *SETD1A/B* mutants (hMUT) through point mutations, which failed to rescue the development of *Setd1a/b* KD embryos (Figure R14E,F). These findings not only exclude potential off-target effects of the *Setd1a/b* siRNAs but also confirm the critical role of SETD1A/B's methyltransferase activity in early embryonic development, distinct from other non-catalytic functions.

As also suggested by Referee 2, the knockdown of *Mll2* did not produce a significant developmental phenotype. Therefore, it is essential to determine whether the knockdown of *Mll2* was effective and sustained, rather than a result of off-target effects. To address this, we initially verified the effective and sustained suppression of MLL2 in early embryos (Figure R15A). Furthermore, in response to your recommendation, we developed three new siRNAs targeting *Mll2*, called *Mll2* KD-2, and carried out independent RNA interference experiments. Upon verifying the knockdown efficiency of these siRNAs, we utilized immunofluorescence staining to assess H3K4me3 and EU levels in *Mll2* KD-2 Late2C embryos (Figure R15B-E). We also observed and quantified early embryonic development following *Mll2* KD-2 (Figure R15F,G). These critical results supported and reinforced our previous findings.

We have added these results in Figure 5H-J, Appendix Figures S11A-B, and Appendix Figures S7 detailed on lines 222-224 and 316-321 of the revised manuscript. We hope that this new evidence will address your concerns.

SETD1A

SETD1B

— SET domain — NSET domain

Figure R13. Alignment of amino acid sequences for SETD1A and SETD1B between human and mouse. Dark blue shades 100% conserved amino acids. The red and blue underlines highlight the regions within the SET and NSET domains that are responsible for histone methyltransferase activity.

Figure R14. The catalytic function of SETD1A/B in early mouse embryonic development. **(A)** Comparison of siRNA1-3 targeting mouse *Setd1a* with the human *SETD1A* coding sequence, and siRNA1-2 targeting mouse *Setd1b* with the human *SETD1B* coding sequence. **(B)** A schematic outlines the methodology for microinjection into MII oocytes, detailing the injection of mouse *Setd1a/b* siRNA to achieve knockdown (*Setd1a/b* KD), the co-injection of *Setd1a/b* siRNA with human WT *SETD1A* and *SETD1B* mRNA (*Setd1a/b* KD + hWT mRNA), and the co-injection of *Setd1a/b* siRNA with human MUT *SETD1A* and *SETD1B* mRNA (*Setd1a/b* KD + hMUT mRNA). **(C)** Immunostaining for SETD1A (red) and DNA (gray) in control, *Setd1a/b* KD, *Setd1a/b* KD + hWT, and *Setd1a/b* KD + hMUT mRNA embryos at the blastocyst stage is shown on the left. The right panel quantifies the relative intensities of blastocyst. Error bars represent SD. n.s., no significance; *** $P < 0.001$; two-sided Student's *t*-test. **(D)** Immunostaining for SETD1B (red) and DNA (gray) in control, *Setd1a/b* KD, *Setd1a/b* KD + hWT, and *Setd1a/b* KD + hMUT mRNA embryos at the blastocyst stage is shown on the left. The right panel quantifies the relative intensities of blastocyst. Error bars represent SD. n.s., no significance; *** $P < 0.001$ two-sided

Student's *t*-test. (E) Images of early embryos from control, *Setd1a/b* KD, *Setd1a/b* KD + hWT, and *Setd1a/b* KD + hMUT mRNA groups, displaying a typical example from three independent experiments. Scale bar, 50 μ m. (F) Line plot showing the development rate of embryos from control, *Setd1a/b* KD, *Setd1a/b* KD + hWT, and *Setd1a/b* KD + hMUT mRNA groups. Error bars represent SD from three biological replicates. ****P* < 0.001; two-sided Student's *t*-test.

Figure R15. Efficient knockdown of *Mll2* in mouse early embryos utilizing three additional siRNAs. (A) Immunostaining for MLL2 (red) and DNA (gray) in control and *Mll2* KD embryos at the Late2C and blastocyst stage is shown on the left. The right panel quantifies the relative intensities of MLL2. Scale bar, 20 μ m. Error bars represent SD. n.s., no significance; ****P* < 0.001; two-sided Student's *t*-test. (B) Bar chart showing the relative expression levels of *Mll2* following *Mll2* KD at the Late2C stage, as detected by RT-qPCR. Error bars represent SD from three biological replicates. ***P* < 0.01; two-sided Student's *t*-test. (C) Immunostaining for MLL2 (red) and DNA (gray) in control and *Mll2* KD-2 embryos at the Late2C and blastocyst stage is

shown on the left. The right panel quantifies the relative intensities of MLL2. Scale bar, 20 μm . Error bars represent SD. *** $P < 0.001$; two-sided Student's t -test. **(D)** Immunostaining for H3K4me3 (green) and DNA (gray) in control and *Mll2* KD-2 embryos at the Late2C stage is shown on the left. The right panel quantifies the relative intensities of H3K4me3. Scale bar, 20 μm . Error bars represent SD. *** $P < 0.001$; two-sided Student's t -test. **(E)** Representative images of the EU-staining for control and *Mll2* KD-2 embryos at the Late2C stages on the left, alongside a quantitative analysis of EU signal intensities on the right panel. Scale bar, 20 μm . Error bars represent SD. n.s., no significance; two-sided Student's t -test. **(F)** Line plots showing the development rate of control and *Mll2* KD-2 embryos at specified time points. Error bars represent SD from three biological replicates. n.s., no significance; two-sided Student's t -test. **(F)** Representative images of control and *Mll2* KD-2 groups at the blastocyst stage. One representative image from three independent experiments is shown. Scale bar, 50 μm .

References:

Hoshii T, Cifani P, Feng Z, Huang CH, Koche R, Chen CW, Delaney CD, Lowe SW, Kentsis A, Armstrong SA (2018) A Non-catalytic Function of SETD1A Regulates Cyclin K and the DNA Damage Response. *Cell* 172: 1007-1021 e17

2. The authors perform their knock-downs in either MII oocytes or zygotes but do not make it clear throughout the manuscript which results arise from MII oocytes and which from zygotes. This makes the work presented extremely difficult to evaluate, especially considering previously published work using conditional and developmental knock-outs of *Mll2*, *Setd1a* and *Setd1b*. Possibly some of the results presented here may confirm, or stand in contradiction to, the previously published work. Consequently, the statement in the abstract "Our research reveals that MLL2 is responsible for depositing H3K4me3 in totipotent embryos, with a transition to SETD1A/B in pluripotent embryos" could either be partially justifiable (it is already known that SETD1A is the major H3K4me3 in pluripotent embryos) or completely unjustified (because it is already known that MLL2 is the major H3K4me3 in late oocytes. Also SETD1B is required for late oogenesis and progress of the zygote). Notably none of these published findings (refs 33, 39, Brici et al, *Development*, 2017) are appropriately discussed because the authors do not compare their results with expectations arising from the existing state of knowledge.

Response: We appreciate the referee's critical observations regarding the lack of clarity in our manuscript concerning the specific stages at which knockdown experiments are performed. In the revised manuscript, we have meticulously ensured that each experiment is accompanied by a corresponding experimental design flowchart. Our study focuses on the role of H3K4me3 during early embryonic development, necessitating the effective knockdown of *Mll2*, *Setd1a*, *Setd1b*, and *Kdm5b* during this critical period. As *Mll2*, *Setd1a*, and *Setd1b* are expressed maternally (Figure R16A,B), initiating knockdown at the zygote stage is insufficient. Consequently, we conducted these experiments in MII oocytes before fertilization. In contrast, *Kdm5a* and *Kdm5b* expression starts at the 2-cell stage (Figure R16C,D), allowing us to begin *Kdm5b* knockdown and CPI-455 treatment from the zygote stage. We hope that the way we present our current results meets your expectations.

Following your advice, we have also compiled a summary of the literature on SET1/MLL complex knockouts and knockdowns in Table R1. Previous research primarily utilized conditional or developmental knockout strategies to study *Mll2*, *Setd1a*, *Setd1b*, and *Cxxc1*. Conditional knockouts of *Mll2*, *Setd1b*, and *Cxxc1* in oocytes markedly affected the quality of oocytes, complicating the assessment of whether the impacts on early embryonic development are direct or secondary to oocyte defects. Developmental knockouts might obscure earlier developmental requirements due to the persistence of maternal mRNA and proteins within the oocytes. For instance, while maternal MLL2, SETD1B, and CXXC1 are crucial for oocyte maturation and fertilization, embryos lacking *Mll2*, *Setd1b*, or *Cxxc1* in zygotes can still develop to the blastocyst stage. Therefore, we targeted maternal reserves by knocking down these genes in MII oocytes to more precisely dissect the role of SET1/MLL complexes in early embryonic development.

In growing oocytes, research indicates that the knockout of *Mll2* or *Cxxc1* results in reduced H3K4me3 levels, whereas the knockout of *Setd1b* does not significantly impact overall H3K4me3 levels. These findings suggest a potential compensatory relationship between SETD1A and SETD1B in H3K4me3 deposition, indicating that the SET1 and MLL2 complexes work collaboratively to regulate H3K4me3 deposition in growing oocytes. Conversely, our data reveal that the MLL2 and SET1 complexes contribute to H3K4me3 deposition at distinct stages of early embryonic development.

In the discussion section (lines 373-387), we have further clarified how our study contrasts with previous reports. We hope this discussion meets your expectations, and we appreciate the opportunity to enhance our manuscript and value your constructive suggestions for the discussion section.

Figure R16. Ribo-seq and mRNA-seq analysis of H3K4me3 modifiers in mouse oocytes and early embryos. **(A-B)** Line graphs illustrate Ribo-seq (RPF) and mRNA-seq (mRNA) levels for *Mll1*, *Mll2*, *Setd1a* and *Setd1b* in mouse oocytes and early embryos, based on Ribo-lite and mRNA-seq data from GSE169632(Zhang *et al*, 2022) **(A)** and GSE165782(Xiong *et al*, 2022) **(B)**. **(C-D)** Line graphs illustrate Ribo-seq (RPF) and mRNA-seq (mRNA) levels for *Kdm5a* and *Kdm5b* in mouse oocytes and early embryos, based on Ribo-lite and mRNA-seq data from GSE169632(Zhang *et al*, 2022) **(C)** and GSE165782(Xiong *et al*, 2022) **(D)**.

Table R1. Summarize research on mouse SET1/MLL complexes knockout or knockdown.

	Reference	Knockdown or knockout	Developmental phenotype	Effect on H3K4me3
Mll2	PMID: 16540515	Developmental knockout	Developmental delay initiated from E7.5, with mortality occurring prior to E11.5	N/A
	PMID: 20808952 PMID: 29323282	Conditional knockout (Gdf9 -Cre)	Oocyte: premature follicle loss, fail to establish transcriptional repression in GV oocyte (SN) Embryo: embryos resulting from cKO♀ x WT♂ matings exhibit developmental arrest between the 1-cell and 2-cell stages	The global level of H3K4me3 in GV oocytes decreases, accompanied by the loss of non-canonical H3K4me3
Setd1a	PMID: 24550110	Developmental knockout	No Setd1a KO embryos were found after E7.5	H3K4me3 is reduced by 50% in E7.5 embryos.
	PMID: 24550110	Conditional knockout (Gdf9 -Cre)	Oocyte: did not perturb fertility Embryo: the knockout phenotype in embryos from cKO♀ x Setd1a ^{FC/+} ♂ is consistent with the developmental knockout	N/A
Setd1b	PMID: 24550110	Developmental knockout	Growth retardation is apparent from E7.5 and embryos die before E11.5	The overall level of H3K4me3 remains unchanged in E10.5 embryos
	PMID: 28619824 PMID: 35137160	Conditional knockout (Gdf9 -Cre)	Oocyte: MII oocytes exhibited irregularities of the zona pellucida and meiotic spindle Embryo: most embryos from cKO♀ x WT♂ crossings fail before the two-cell stage	H3K4me3 levels remain unchanged in cKO oocytes, but it is redistributed across the genome
Cxxc1	PMID: 11604496	Developmental knockout	Embryonic lethality as being between 3.5 and 6.5 dpc	N/A
	PMID: 28768200	Conditional knockout (Zp3 -Cre)	Oocyte: failed to complete maturation, overall transcriptional activity was reduced in growing oocyte Embryo: embryos from cKO♀ x WT♂ crossings fail to develop beyond the 2-cell stage	Reduced H3K4me3 in oocytes and zygotes
	PMID: 31676962 PMID: 33621320	Conditional knockout (Gdf9 -Cre)	Both follicle growth and ovulation were impaired, with reduced transcriptional activity in growing follicles	H3K4me3 levels were broadly down-regulated across canonical and non-canonical peaks
Wdr82	PMID: 21123813	Knockdown (Microinjection with siRNA or antibody)	Blastocyst formation rate decreased	H3K4me3 levels decreased in 4-cell and morula embryos but remained unchanged in

				2-cell embryos
--	--	--	--	----------------

The manuscript includes various misunderstandings.

a) line 62, in their demonstration that MLL2 is responsible for most H3K4me3 deposition on bivalent promoters in ESCs, Denissov et al (ref 34) also showed that MLL2 is found on most active (as well as bivalent) promoters. So the statement "whereas MLL1 and MLL2 target developmental gene promoters in a gene specific manner 15,16" is not correct. Ref 16 does not claim to show that MLL1 targets developmental gene promoters in a gene specific manner. MLL1, as does MLL2, appears to bind to most active promoters e.g. Guenther et al, PNAS 102, 8603 (2005).

Response: Thank you for your careful review and insightful comments. As you highlighted, MLL1 and MLL2 bind to the majority of active promoters, which do not depend on MLL1 and MLL2 for H3K4me3 deposition. Therefore, we have amended the sentence to: "SETD1A and SETD1B predominantly mediate global H3K4me3 deposition (Clouaire et al, 2012; Sze et al, 2020), whereas MLL1 and MLL2 selectively catalyze H3K4me3 at specific gene promoters (Denissov et al, 2014; Hu et al, 2013a; Wang et al, 2009)" as stated in the revised manuscript (Lines 64-66), and have corrected the appropriate references.

b) line 72-74. "Previous research indicates that the deficiency of any histone lysine methyltransferase results in embryonic fatality between E7.5 and E11.5 12" is not correct. Presumably this statement refers to H3K4 lysine methyltransferases and not all the other lysine methyltransferase as well. Even then it is not correct. *Setd1a* knock-outs stop at E5.5 and do not die at E7.5 (ref 39) and MLL3 knock-outs proceed to birth (Jang et al, 2013).

Response: We are grateful for the referee's corrections and guidance. We recognize the inaccuracy in our previous description concerning the timing of embryonic lethality due to the loss of histone lysine methyltransferases. We have amended the sentence to: "Previous research indicates that embryos lacking the H3K4 methyltransferases SETD1A and MLL2 are non-viable before E11.5 (Bledau, Schmidt et al., 2014, Glaser, Schaft et al., 2006). In contrast, embryos with a *Setd1b* deletion survive beyond E11.5, while the viability of *Mll1*-deficient embryos ranges from the 2-cell stage to E14.5, depending on the knockout strategy used (Bledau et al., 2014, Cenik & Shilatifard, 2021)." as stated in the revised manuscript (Lines 77-81).

c) Figure 1B, presumably the 55,335 peaks refers to all H3K4me3 peaks and not 'All genes'. The mouse genome has 22,000 genes and about half are expressed in early development, so the figure legend (line 140) 'Each row represents a promoter region (TSS +/- 2 kb) cannot be correct.

Response: Thank you for your attention to this detail. We have revised the figure and its legend to reflect 22,470 genes.

c) line 185/6 and Figure 2D. 'Significantly, transcriptional inhibition selectively reduced H3K4me3 in morulae on shared peaks ...' - is very hard to see on Fig. 2D.

Response: Thank you for pointing out the inaccuracies in our manuscript regarding the description of the results. We have revised the description as follows: "On the shared peaks of

H3K4me3 between Late2C embryos and morulae, treatment with a transcriptional inhibitor significantly reduced the levels of H3K4me3 in morulae, while the levels of H3K4me3 in Late2C embryos remained unaffected.” as stated in the revised manuscript (Lines 185-188). We hope this revised description more accurately reflects the results of our experiments.

d) line 199, Figure 2 legend. 'PolIII52' - there is no reference 52.

Response: We sincerely apologize for this. We have used the correct citation in the revised manuscript.

e) line 451 - 456. 'Despite initial findings linking ncH3K4me3 in mouse oocytes and pre-ZGA embryos with transcriptional silence, its role in in transcriptional suppression remains contentious.....However, later analyses using H3K4me3 ChIP-seq and RNA-seq knockout oocytes showed that the lack of ncH3K4me3 doesn't directly impede transcription10'. These statements are not correct. First, the linkage between ncH3K4me3 and transcriptional silencing is not contentious (however the molecular explanation remains open). Second, the 'later analyses showed that the lack of showed that the lack of ncH3K4me3 doesn't directly impede transcription10' doesn't directly impede transcription10'. But this has nothing to do with global transcriptional silencing which correlates with ncH3K4me3. This confusing illogicality is used by the authors to establish unjustified priority for their findings. It is followed in the Discussion by a series of unreasonable contentions (lines 458 - 462).

Response: We appreciate the referee's insightful comments and apologize for the oversights in our manuscript. We recognize that the correlation between ncH3K4me3 and transcriptional repression is well-established, and RNA-seq data may not capture the full scope of transcriptional activity. Accordingly, we have updated the manuscript as follows: “Preliminary immunostaining studies have shown that either the knockout of *Mll2* or overexpression of *Kdm5b* leads to reduced levels of ncH3K4me3, which disrupts genomic silencing in mature oocytes(Andreu-Vieyra *et al.*, 2010; Zhang *et al.*, 2016). These results establish a correlation between ncH3K4me3 in mouse oocytes and pre-ZGA embryos and the regulation of transcriptional silencing in mouse oocytes and pre-ZGA embryos. Interestingly, we found that MLL2 deposits ncH3K4me3 in embryos pre-ZGA and cH3K4me3 during ZGA, with neither directly influencing ZGA gene transcription. In ESCs, MLL2 establishes H3K4me3 at the promoters of developmental genes, protecting them from repression by PRC2-derived H3K27me3 or DNA methylation, rather than directly regulating transcription(Douillet *et al.*, 2020). Similarly, in mature mouse oocytes, ncH3K4me3 is predominantly observed in regions of DNA hypomethylation and exhibits mutual exclusivity with H3K27me3(Xu *et al.*, 2019; Zhang *et al.*, 2016; Zheng *et al.*, 2016). This suggests that MLL2's role in establishing H3K4me3 may indirectly affect transcription by modulating other epigenetic markers such as H3K27me3 and DNA methylation, rather than through direct transcriptional regulation. The extensive loss of H3K27me3 and DNA methylation in early mouse embryos (Liu *et al.*, 2016; Smith *et al.*, 2012; Zheng *et al.*, 2016) may explain why reductions in MLL2-mediated H3K4me3 do not influence ZGA gene transcription.” (revised manuscript, Lines 394-404).

f) line 462 - 464 'The absence of H3K27me3 and DNA methylation in early embryos indicates that MLL2's role in early embryogenesis is distinct from its function in ESCs'. This statement is not only illogical (how does 'the absence of H3K27me3 and DNA methylation in early embryos' indicate anything about MLL2?) but also fails to acknowledge that it has already been published that 'MLL2's role in early embryogenesis is distinct from its function in ESCs' (and not cited).

Response: As previously noted, we have thoroughly revised the logic throughout this section. We hope these amendments address your concerns and meet your expectations.

Additional comments -

1. The authors generally equate loss of MLL2 or SETD1A/B to reduced H3K4me3 without considering that these very large proteins are embedded in very large complexes of at least 8 other proteins. They are not simply H3K4 methyltransferase and all these proteins have many other functions that these knock-downs also perturb.

Response: We appreciate the referee's insights regarding the multifaceted roles of MLL2 and SETD1A/B within larger protein complexes. We recognize that these proteins are implicated in H3K4 methylation but also engage in a variety of other functional activities that could be influenced by the knockdowns. To elucidate the specific roles of SETD1A and SETD1B in early embryonic development, we engineered catalytically inactive mutants via point mutations and subsequently overexpressed both wild-type and mutant mRNAs following the knockdown of *Setd1a/b* (Figure R14). Our findings revealed that only the wild-type SETD1A and SETD1B mRNAs could ameliorate the developmental defects, thus confirming that the influence of SETD1A/B on early embryogenesis is predominantly mediated through their H3K4 methyltransferase activity. Notably, the knockdown of *Mll2* did not result in significant developmental anomalies in early embryogenesis, thereby there is no need to concern about its non-catalytical function.

2. The authors perpetuate the confusion accompanying the term 'COMPASS', which means Complex associated with Set1. Mammals have six H3K4 methyltransferase complexes only two of which are associated with SET1. The other four mammalian complexes have substantial differences. The core problem here is that the term 'COMPASS' is imprecise. It is not correct to call all H3K4 methyltransferase complexes 'COMPASS' in the same way it would not be correct to call MLL1, MLL2, MLL3 or MLL4 'SET1'. Line 59 is a relevant example of this confusion where the six separate mammalian complexes are bundled together under the term 'COMPASS complex'. Whereas insiders in this field are used to this problem, there is no justifiable reason to keep perpetuating this confusion. I recommend 'SET1/MLL complexes', which has utility and is more precise.

Response: We appreciate the referee's insightful comment regarding the incorrect use of the term "COMPASS" in our manuscript. We have amended our terminology to "SET1/MLL complexes" throughout the document to ensure accuracy and to prevent any further confusion.

Dear Dr. Miao,

Thank you for submitting a revised version of your manuscript. We have now received input from two of the original reviewers, who find that most of previous concerns have been addressed satisfactorily and recommend acceptance of the manuscript after a minor revision as requested by reviewer #2.

Additionally, there remain a few editorial points that need addressing before I can extend official acceptance of the manuscript:

1. In the Author Checklist, please fill the column E for rows 69 (core facilities) and 115 (data citations).
2. Please merge the Data Availability and Code Availability sections. I could not access the dataset deposited in Genome Sequence Archive via the provided URL, please check. The code should also be made publicly available via an appropriate depository or database, and an URL for direct access should be provided.
3. CRedit has replaced the traditional author contributions section because it offers a systematic, machine-readable author contributions format that allows for more effective research assessment. Please remove the Authors Contributions from the manuscript and use the free text boxes beneath each contributing author's name in our online submission system to add specific details on the author's contribution. More information is available in our guide to authors.
4. Please rename "Declaration of interests" section into "Disclosure and competing interests statement" (further info: <https://www.embopress.org/page/journal/14602075/authorguide#conflictsofinterest>).
5. Figure panel 1B is not mentioned in the manuscript text, please add the appropriate callout.
6. Source data files need to be saved in a scheme one figure/folder and then uploaded as .zip files. All source data files for main figures need to be saved in a single folder per figure and this needs to be zipped and then uploaded labelled as, for example, "SD figure 1.zip" file. For EV and/or appendix figures, please ZIP together all source data in a single folder.
7. Our data editors have flagged the following issues in figure legends that need correcting:
 - Please provide the exact p values in the legends of figures 4C; 5A, E, G, J; 6C, D, E, G.
 - In figure 6D there is a mismatch between the annotated p values in the figure legend and the annotated p values in the figure file that should be corrected.
 - Please note that the p value is not represented in the figure 3B, H, 4F, 5D, 6F, however statistical test related information is provided in the legend of the corresponding figure. This needs to be rectified.
 - Please define the box plots in terms of minima, maxima, centre, bounds of box and whiskers, and percentile in the legends of figures 4C, H; 5E, G; 6E, G.
 - Please provide information on the number and nature of replicates in the legends of figures 4C, H; 5G, 6E, G.
 - Please define the measure of center for the error bars in the legends of figures 5A, J; 6C, D
 - Please provide a numbered scale bar for the heatmap present in figures 1C, D; 2A.
8. Please submit the synopsis image as a separate image file (jpg or png format) with the dimensions of 500 pixels width - 300-600 pixels height.

With best wishes,

Ieva

Ieva Gailite, PhD
Senior Scientific Editor
The EMBO Journal
Meyerohofstrasse 1
D-69117 Heidelberg
Tel: +4962218891309
i.gailite@embojournal.org

We realize that it is difficult to revise to a specific deadline. In the interest of protecting the conceptual advance provided by the work, we recommend a revision within 3 months (6th Feb 2025). Please discuss the revision progress ahead of this time with the editor if you require more time to complete the revisions.

Referee #1:

The authors appropriately responded to the comments from the reviewers.
I appreciate the authors' great efforts to revise the manuscript and support the acceptance of this paper in EMBO J.

Referee #2:

The authors have provided important new results and clarification to address the issues raised by me and the other two reviewers. Critical new results include demonstration of the efficacy of the siRNA knock-downs of MLL2, SETD1A or SETD1B by immunofluorescence analysis of embryos: this corrects the major omission of the original manuscript, which had almost no indication at the protein level of the effects of the knock-downs. Importantly, also, the IF and RNA-seq analysis indicate the persistence of the knock-down effects across the developmental stages examined. The authors have also rephrased some statements, such as referring to the effects of the knock-downs as 'reduction' rather than 'loss'. In their response to my Key Question 1, they comment "Mll2 KD results in a substantial decrease (exceeding 75%) in both mRNA and protein levels of MLL2 in early embryos, as shown in Figure R7". I believe they do not make a claim in the actual Results of the degree of protein knock-down achieved, and I think this is correct. They present quantification of IF intensity (e.g., Appendix Figure S1), which is there for readers to evaluate. But precise claims of the exact degree of knock-down would be difficult to substantiate because of uncertainty of the linearity of the IF signals.

In response to a point raised by Reviewer 3, the authors have also repeated the MLL2 knock-down by independent siRNAs, to mitigate against off-target effects, as well as using rescue experiments, e.g., by expression of WT or MUT SETD1A/B.

Together, these responses represent substantial additional analysis/experiments undertaken by the authors to validate the efficacy of the siRNA knock-downs.

Other points:

Lines 65-66: the statement "whereas MLL1 and MLL2 selectively catalyse H3K4me3 at specific gene promoters" seems still to be incorrect, as Reviewer 3 has pointed out in their comments that MLL1 and MLL2 appear to bind to most active promoters. I think the authors have misunderstood the correction the expert reviewer has recommended.

Line 219: the number of up- and down-regulated genes referred to in the text (225, 368) do not agree with the numbers in the cited figure (Appendix Fig. S6E: 224 and 369).

There is a similar discrepancy on Line 270 in referring to Figure 4F (147 instead of 145).

The authors are encouraged to carefully check these numbers and in other cases where they are referencing values in figures.

Response to Referee #1:

The authors appropriately responded to the comments from the reviewers.

I appreciate the authors' great efforts to revise the manuscript and support the acceptance of this paper in EMBO J.

Response: Thank you.

Response to Referee #2:

The authors have provided important new results and clarification to address the issues raised by me and the other two reviewers. Critical new results include demonstration of the efficacy of the siRNA knock-downs of MLL2, SETD1A or SETD1B by immunofluorescence analysis of embryos: this corrects the major omission of the original manuscript, which had almost no indication at the protein level of the effects of the knock-downs. Importantly, also, the IF and RNA-seq analysis indicate the persistence of the knock-down effects across the developmental stages examined. The authors have also rephrased some statements, such as referring to the effects of the knock-downs as 'reduction' rather than 'loss'.

In their response to my Key Question 1, they comment "MLL2 KD results in a substantial decrease (exceeding 75%) in both mRNA and protein levels of MLL2 in early embryos, as shown in Figure R7". I believe they do not make a claim in the actual Results of the degree of protein knock-down achieved, and I think this is correct. They present quantification of IF intensity (e.g., Appendix Figure S1), which is there for readers to evaluate. But precise claims of the exact degree of knock-down would be difficult to substantiate because of uncertainty of the linearity of the IF signals.

In response to a point raised by Reviewer 3, the authors have also repeated the MLL2 knock-down by independent siRNAs, to mitigate against off-target effects, as well as using rescue experiments, e.g., by expression of WT or MUT SETD1A/B.

Together, these responses represent substantial additional analysis/experiments undertaken by the authors to validate the efficacy of the siRNA knock-downs.

Response: We appreciate the reviewer's recognition of our work.

Other points:

Lines 65-66: the statement "whereas MLL1 and MLL2 selectively catalyse H3K4me3 at specific gene promoters" seems still to be incorrect, as Reviewer 3 has pointed out in their comments that MLL1 and MLL2 appear to bind to most active promoters. I think the authors have misunderstood the correction the expert reviewer has recommended.

Response: We thank the reviewer for pointing out our mistakes, and we have made the corrections in the manuscript. We have revised the statement to "whereas MLL1 and MLL2 selectively catalyse H3K4me3 at most active gene promoters" in line 65-66.

Line 219: the number of up- and down-regulated genes referred to in the text (225, 368) do not agree with the numbers in the cited figure (Appendix Fig. S6E: 224 and 369).

Response: We apologize for the mistake and have made the revisions in the manuscript.

There is a similar discrepancy on Line 270 in referring to Figure 4F (147 instead of 145).

Response: We sincerely apologize for this mistake and have made the correction in the manuscript.

The authors are encouraged to carefully check these numbers and in other cases where they are referencing values in figures.

Response: Thank you to the reviewer for this suggestion. We have reviewed the manuscript and corrected the mistakes.

Dear Dr. Liu,

Thank you for addressing the final editorial points. I sincerely apologise for the delay in communicating the decision due to conference travel. I am now pleased to inform you that your manuscript has been accepted for publication.

Before we forward your manuscript to our publishers, I would like to propose some edits in the manuscript title, abstract and synopsis (please see below and the attached text file). I have also written a short blurb that will accompany the title of your manuscript in our online system. Please let me know if any corrections or adjustments are needed.

Alternative title:

Histone methyltransferases MLL2 and SETD1A/B play distinct roles in H3K4me3 deposition during the transition from totipotency to pluripotency

Blurb:

SETD1A/B-dependent histone H3 lysine 4 trimethylation at active promoters is required for the first cell fate transition in mouse early embryogenesis.

Synopsis:

During mammalian zygotic genome activation, a shift occurs from non-canonical histone H3 lysine 4 trimethylation (H3K4me3) linked to transcriptional repression to canonical H3K4me3 indicating active promoters. This study reveals that early embryos undergo the transition from totipotency to pluripotency by employing a relay between histone methyltransferases MLL2 and SETD1A/B to deposit H3K4me3.

- MLL2-dependent canonical H3K4me3 deposition occurs in a transcription-independent manner.
- SETD1A/B maintained canonical H3K4me3 distribution in a transcription-dependent manner.
- Disruption of MLL2-mediated H3K4me3 deposition does not affect zygotic genome activation.
- Transcription-dependent H3K4me3 catalyzed by SETD1A/B is essential for the first cell fate decision.

Finally, we would like to promote your manuscript among the Chinese readership. Therefore, we would like to invite you to prepare a short summary of the manuscript in Chinese (1500-2000 Chinese characters), which we will promote on the WeChat platform 'BioArt' with more than 610,000 followers.

If you are interested in this opportunity, we recommend covering the article very close to its online publication date. Thus, ideally we would very much appreciate if you could send us a draft within the next 7 working days. Please let us know whether or not you would be interested in contributing such a short summary in Chinese.

I have included below some general guidelines on how to prepare a summary and a link to recent examples for your reference. Please let me know if you have any questions about this.

If you have any questions, please do not hesitate to contact the Editorial Office. Thank you for this contribution to The EMBO Journal and congratulations on a nice study!

With best wishes,

leva

leva Gailite, PhD
Senior Scientific Editor
The EMBO Journal
Meyerhofstrasse 1

D-69117 Heidelberg
Tel: +4962218891309
i.gailite@embojournal.org

General WeChat Summary Guidelines

1. These summary articles are meant to be targeting general audience so please limit the use of specialized technical terms, acronyms and jargon.
2. A summary usually starts with brief background information of the reported work, which is followed by explaining the findings in some detail, and ends with a short review of the conclusions as well as the implications of the work and future directions for the research.
3. The summary should at least contain one graphical item, such as a scheme or a figure from the paper.
4. Please provide ONE SINGLE document containing all text and graphical materials, ideally as a Word.docx or .doc file. Please DO NOT provide the document as a .pdf file.
5. Please DO NOT publicly release the document before the paper is officially published online.

Summary Examples

EMBO J | 罗招庆/欧阳松应揭示谷酰胺脱氨酶MvcA的去泛素化功能

EMBO J | 王松灵院士团队揭示组织内应力调控大型哺乳动物乳恒牙替换的新机制
